# Chromosomal instability induced in cancer can enhance macrophage-initiated immune responses that include anti-tumor IgG

**Brandon H Hayes**[1,2,3], **Mai Wang**[1,2], **Hui Zhu**[1,2], **Steven H Phan**[1,2], **Lawrence J Dooling**[1,2], **Jason C Andrechak**[1,2,3], **Alexander H Chang**[1,2], **Michael P Tobin**[1,2,3], **Nicholas M Ontko**[1,2], **Tristan Marchena**[1,2], **Dennis E Discher**[1,2,3]*

[1]Physical Sciences Oncology Center at Penn, University of Pennsylvania, Philadelphhia, United States; [2]Molecular and Cell Biophysics Lab, University of Pennsylvania, Philadelphia, United States; [3]Bioengineering Graduate Group, University of Pennsylvania, Philadelphia, United States

*For correspondence:
discher@seas.upenn.edu

Competing interest: The authors declare that no competing interests exist.

**Abstract** Solid tumors generally exhibit chromosome copy number variation, which is typically caused by chromosomal instability (CIN) in mitosis. The resulting aneuploidy can drive evolution and associates with poor prognosis in various cancer types as well as poor response to T-cell checkpoint blockade in melanoma. Macrophages and the SIRPα-CD47 checkpoint are understudied in such contexts. Here, CIN is induced in poorly immunogenic B16F10 mouse melanoma cells using spindle assembly checkpoint MPS1 inhibitors that generate persistent micronuclei and diverse aneuploidy while skewing macrophages toward a tumoricidal 'M1-like' phenotype based on markers and short-term anti-tumor studies. Mice bearing CIN-afflicted tumors with wild-type CD47 levels succumb similar to controls, but long-term survival is maximized by SIRPα blockade on adoptively transferred myeloid cells plus anti-tumor monoclonal IgG. Such cells are the initiating effector cells, and survivors make de novo anti-cancer IgG that not only promote phagocytosis of CD47-null cells but also suppress tumor growth. CIN does not affect the IgG response, but pairing CIN with maximal macrophage anti-cancer activity increases durable cures that possess a vaccination-like response against recurrence.

## eLife assessment

The authors provide **compelling** evidence that MSP1 inhibition (leading to chromosomal instability or CIN in the cancer cells) increases phagocytosis and that tumors with CIN respond better to macrophage therapeutics. In this **important** study, they demonstrate particularly impressive survival rates for mouse models of CIN B16 tumors treated with adoptively transferred macrophages, CD47-SIRPα blockade, and anti-Tyrp1 IgG.

## Introduction

Chromosomal instability (CIN) has long been associated with poor prognosis and reduced immune cell activity against tumors (*Davoli et al., 2017*; *Vasudevan et al., 2021*). The high frequency of chromosome mis-segregation in CIN often generates micronuclei and can cause aneuploidy – an abnormal ratio of chromosomes. CIN-induced genetic heterogeneity can serve as a tumor promotor (*Sheltzer et al., 2017*) and allow some tumor subpopulations to favor aggression, metastatic potential, immune

evasion, and resistance to therapies (*Ben-David and Amon, 2020*; *Chunduri and Storchová, 2019*; *Vasudevan et al., 2021*). However, early-stage CIN also induces anti-cancer vulnerabilities (*Cohen-Sharir et al., 2021*; *Vasudevan et al., 2020*), such as proliferation deficits (*Wang et al., 2021*). Early-stage CIN-afflicted cells have yet to adapt and achieve aneuploidies that favor growth and immune evasion (*Tripathi et al., 2019*; *Vasudevan et al., 2021*). CIN in diploid cells that is caused by spindle assembly checkpoint disruption via MPS1 kinase inhibition (MPS1i) further induces a senescence-associated secretory pathway phenotype and upregulation of NF-κB and interferon-mediated pathways, among others, that drive immune clearance of chromosomally aberrant cells (*Santaguida et al., 2017*; *Wang et al., 2021*). CIN and ploidy changes also inhibit tumor growth in immunocompetent mice while having little effect in immunocompromised mice (*Senovilla et al., 2012*; *Boilève et al., 2013*), which suggests CIN somehow increases immunogenicity.

Recent analyses of The Cancer Genome Atlas (TCGA) showed that highly aneuploid tumors include macrophages that are polarized toward a pro-cancer, M2-like phenotype, among other pro-cancer immune changes (*Davoli et al., 2017*; *Taylor et al., 2018*). MPS1i-treated cancer cells have also been reported to escape immune-mediated clearance (*Wang et al., 2021*), with another study reporting that cancer cells respond to CIN with IL6-STAT3 signaling (*Hong et al., 2022*) that protects from CIN-induced cell death, allows cells to adapt to CIN- and aneuploidy-induced stresses, and minimizes interferon-related anti-cancer responses. Pro-survival signaling amidst CIN also increases factors (such as IL6) that can induce a pro-cancer, M2-like phenotype (*Fernando et al., 2014*). On the other hand, ploidy changes tend to increase factors that can promote macrophage-mediated phagocytosis (*Chao et al., 2010*; *Krysko et al., 2018*), which led us to hypothesize that macrophages can be manipulated to more productively counter CIN- and aneuploidy-afflicted tumors.

Phagocytosis of 'self' cells including cancer cells is generally inhibited by the macrophage checkpoint interaction between SIRPα on the macrophage and ubiquitously expressed CD47 on any target cell (*Oldenborg et al., 2000*; *Tsai and Discher, 2008*). While tumor cell engulfment can be driven to some extent via IgG opsonization by using anti-tumor monoclonal antibodies that bind Fc receptors on macrophages (*Alvey et al., 2017*; *Suter et al., 2021*), this is generally insufficient to eliminate cancers, especially solid tumors. Maximal macrophage-mediated phagocytosis is achieved when CD47-SIRPα signaling is simultaneously disrupted. However, achieving tumor rejection is still a major challenge for macrophage-oriented therapies in clinically relevant immunocompetent mice with syngeneic tumors (*Ingram et al., 2017*; *Sockolosky et al., 2016*). In patients, multiple clinical trials of antibody blockade of CD47 have been stopped due to safety concerns and little to no efficacy - which motivates more fundamental studies as well as safer approaches. Additionally, macrophages often polarize toward tumor-associated macrophage (TAM) phenotypes that tend to correlate with poor clinical prognoses (*Cerezo-Wallis et al., 2020*; *Noy and Pollard, 2014*), have poor phagocytic function, and can promote tumor growth (*Georgouli et al., 2019*). The cited TCGA analyses that indicated pro-cancer macrophages predominate in highly aneuploid tumors almost certainly reflects selection for cancer growth but might not reflect high levels of ongoing CIN.

We hypothesized that early-stage CIN in cancer cells can stimulate anti-cancer activity of macrophages sufficient to dominate proliferation of high CIN tumors, but probably only when maximizing pro-phagocytic conditions, such as with blockade of SIRPα (rather than ubiquitous CD47). Our results provide initial in vitro and in vivo evidence that CIN indeed favors anti-cancer macrophage activity and consistently increases durable cures in mice under conditions of maximal phagocytosis. Elimination of these CIN-afflicted solid tumors further drives development of both anti-cancer opsonizing IgGs and enhanced cell-mediated immunity, both of which can help suppress growth against aggressive chromosomally stable tumors.

## Results
### CIN-afflicted tumors skew macrophages toward an anti-cancer phenotype

To investigate the possible effects that cancer cell CIN may have in mediating macrophage immune response, we focused on the poorly immunogenic B16F10 mouse melanoma model. To induce CIN, we treated cells for 24 hr with the MPS1 inhibitor reversine (*Hong et al., 2022*; *Kitajima et al., 2022*; *Santaguida et al., 2017*; *Santaguida et al., 2010*), and after washout and recovery for 48 hr

(*Figure 1A*), we quantified CIN in B16F10 cells, characterized its effects in early-stage tumors, and studied potential effects on macrophages in vitro (*Figure 1A*). MPS1i increased the frequency of micronuclei visible in interphase cells (*Figure 1B*), as micronuclei are often used as a surrogate for CIN and genome instability (*Cohen-Sharir et al., 2021*; *Crasta et al., 2012*; *Harding et al., 2017*; *Mackenzie et al., 2017*). Regardless of the MPS1i concentration, we saw >10-fold increases of micronuclei over the cell line's low basal level (~1% of cells), and two other MPS1i inhibitors AZ3146 and BAY12-17389 confirm such effects (*Figure 1—figure supplement 1A*). Micronuclei-positive cells can persist up to 12 days after treatment (*Figure 1—figure supplement 1B*), while control cells maintain the low basal levels. The results suggest pre-treatment with MPS1i can simulate CIN in experimental contexts even for 1–2 weeks.

Upon confirmation of increased micronuclei induction, we then proceeded to quantitatively assess copy number variations that resulted from MPS1i treatment using single-cell RNA-sequencing. For DMSO-treated B16F10 cells, approximately 10% of the population were considered aneuploid (*Figure 1C – top*; *Figure 1—figure supplement 1C*) – cells with copy number profiles that are 2.5 standard deviations away from the distribution peak (see Materials and methods). In comparison, 34% of the MPS1i-treated B16F10 cells were aneuploid (*Figure 1C – bottom*; *Figure 1—figure supplement 1C and D*).

Given that MPS1i induces CIN and quantifiable aneuploidy in B16F10 cells, we next sought to assess whether CIN may affect early-stage immune response and tumor development. We proceeded to establish tumors in mice with either MPS1i-treated or DMSO-treated B16F10 (following the same schema outlined in *Figure 1A*). We isolated tumors from mice at two timepoints, 5 and 10 days after initial challenge, and then we processed whole tumors for bulk RNA-sequencing (*Figure 1D*). Comparison of tumors comprised of MPS1i-treated and DMSO-treated cells by differential gene expression showed distinct downregulated transcripts for numerous genes encoding M2-like (pro-cancer) macrophage polarization markers (*Figure 1E*). At day 5 post-challenge, classical M2 markers *Arg1*, *Marco*, and *Cd274* are downregulated, and although several M2 markers are upregulated (*Mrc1*, *Cd163*, and C1q complement genes), by day 10, these, *Marco*, and *Cd27* are all downregulated. Furthermore, at both days 5 and 10, cytokines *Ccl2*, *Ccl4*, *Ccl7*, *Ccl22*, and *Il21r* associated with an M2 phenotype (*Cerezo-Wallis et al., 2020*) were found to be consistently downregulated. Gene set enrichment analysis (GSEA) further revealed that pathways related to DNA damage, cell cycle, and growth were also downregulated (*Figure 1F*), consistent with previous studies (*Cohen-Sharir et al., 2021*; *Wang et al., 2021*). These downregulated pathways align with CIN-associated checkpoints on cell growth.

Downregulation of M2 macrophage markers and cytokines in CIN-afflicted tumors in vivo led us to address whether CIN in cancer cells might influence macrophages via the secretome, as suggested from previous in vitro studies of bone marrow-derived macrophages (BMDMs) (*Cerezo-Wallis et al., 2020*; *Xian et al., 2021*). Conditioned media from MPS1i-treated or DMSO-treated B16F10 cells was collected and added for 24 hr to differentiated BMDMs. BMDMs were then processed for single-cell RNA-sequencing (*Figure 1G*), and dimensionality reduction of gene expression via uniform manifold approximation and projection (UMAP) identified four distinct macrophage population clusters. Two clusters (0 and 2) consisted of ~75% of macrophages treated with MPS1i-treated B16F10 secretome, and two clusters (1 and 3) consisted of ~75% of macrophages treated with DMSO-treated B16F10 secretome (*Figure 1H*). The latter showed increased expression of M2-like pro-cancer and anti-inflammatory markers (*Figure 1I*). Consistent with the whole-tumor bulk RNA-sequencing, expression was increased for *Mrc1* in both clusters 1 and 3 and for *Ccl2*, *Ccl4*, and *Ccl7* in cluster 3. More importantly though, these same clusters 1 and 3 tended to downregulate M1-like anti-cancer and pro-inflammatory genes (*Figure 1J*). Clusters 0 and 2 that consisted mostly of MPS1i-treated B16F10 secretome showed increased expression of pro-inflammatory, M1-like anti-cancer markers and little-to-no expression of most M2-like, pro-cancer markers (*Figure 1I and J*). These results suggest that early-stage CIN in cancer cells can activate macrophages via secreted factors toward M1-like anti-cancer activity.

## CIN-afflicted, CD47-knockout tumoroids are eliminated by macrophages

To assess functional effects of macrophage polarization, we focused on a three-dimensional (3D) 'immuno-tumoroid' model in which macrophage activity can work (or not) over many days against

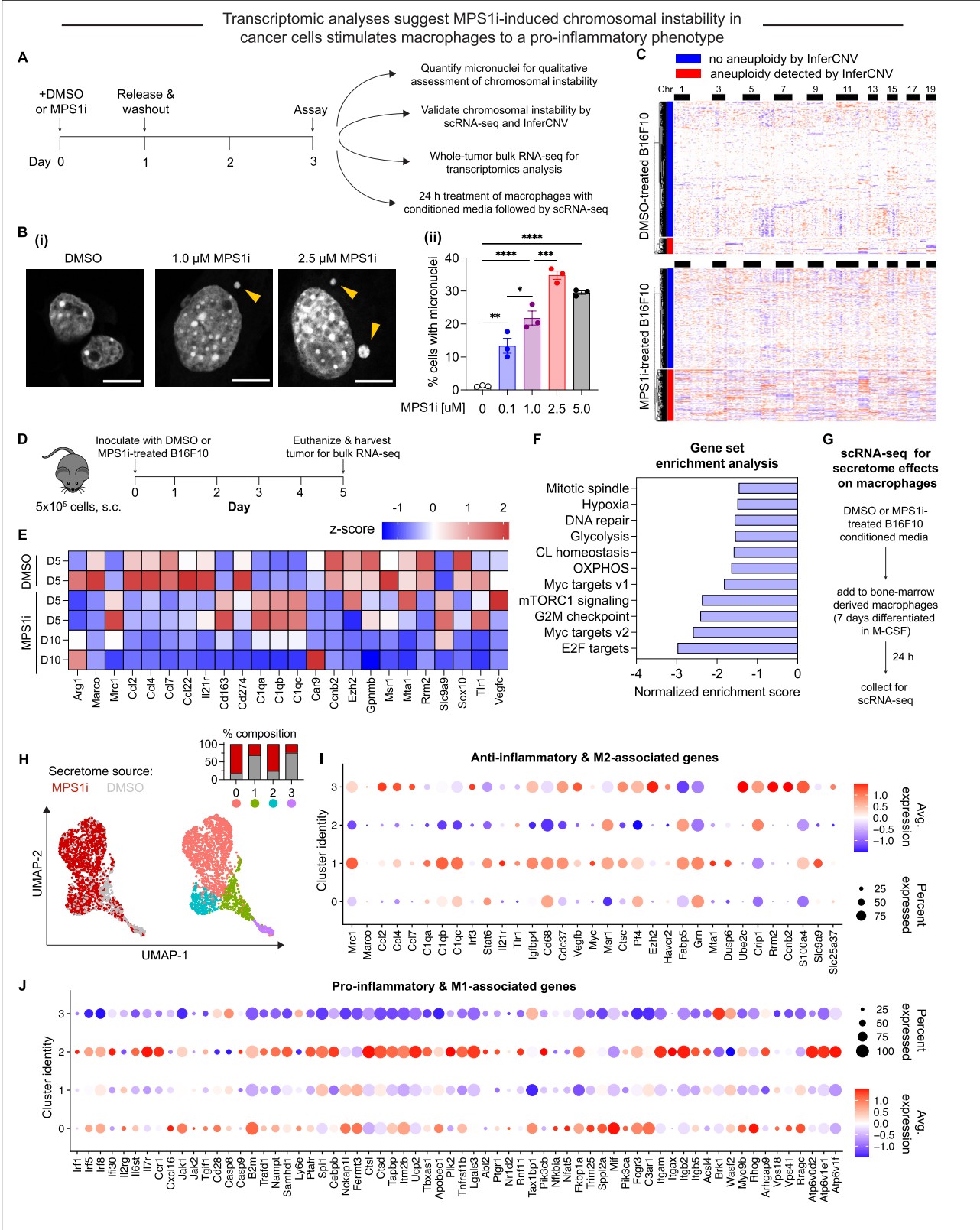

**Figure 1.** MPS1 kinase inhibition (MPS1i)-induced chromosomal instability generates microenvironment conditions that skew toward anti-cancer M1-like macrophages relative to a pro-cancer M2-like phenotype. (**A**) Schematic for treatment of B16F10 mouse melanoma cells with MPS1 inhibitors (MPS1i), such as reversine, or the equivalent volume of DMSO vehicle control. After 24 hr, cells were washed twice with phosphate-buffered saline (PBS) and allowed to recover for an additional 48 hr. Follow-up experiments assessed chromosomal instability, aneuploidy, and effects on bone marrow-

*Figure 1 continued on next page*

*Figure 1 continued*

derived macrophages (BMDMs). (**B**) (**i**) Representative DNA images of B16F10 cells after treatment. Scale bars = 10 μm. Yellow arrowheads point at micronuclei, which signify chromosomal instability. (**ii**) Quantification of the percentage of cells with micronuclei across a range of different MPS1i concentrations (mean ± SEM shown, n = 3 replicates per condition). Statistical significance was calculated by ordinary one-way ANOVA and Tukey's multiple comparison test (*p<0.05; **p<0.01; ***p<0.001; ****p<0.0001). (**C**) Inferred copy number in DMSO and MPS1i-treated B16F10 cells from single-cell RNA-sequencing and the InferCNV pipeline. Cells that are considered aneuploid (labeled as outliers in the InferCNV algorithm) show full-level chromosome gains and/or losses compared to the consensus copy number profile for most DMSO-treated cells. 34% of MPS1i-treated cells show aneuploidy, as determined by InferCNV, compared to 10% of DMSO-treated cells. (**D**) Schematic for whole-tumor bulk RNA-sequencing. Prior to tumor inoculation, CD47 knockout (KO) B16F10 cells were treated with 2.5 μM MPS1i (reversine) or the equivalent volume of DMSO vehicle control. Cells were treated for 24 hr, after which they were washed twice with PBS and allowed to recover for 48 hr. C57BL/6 mice were then subcutaneously injected with $5 \times 10^5$ cells and euthanized 5 or 10 days later, with tumors excised and disaggregated for sequencing. (**E**) Heatmap of selected RNA transcripts related to M2 macrophage polarization and pro-tumor function that were differentially expressed in tumors of MPS1i-treated B16F10 compared to DMSO pre-treated cells. Heatmap shows $\log_2$-transformed transcript reads as z-score normalized. Overall, M2 markers are downregulated at day 5 and even further at day 10 compared to DMSO controls. Tumors analyzed: two comprised of DMSO-treated cells 5 days post-inoculation, two comprised of MPS1i-treated cells 5 days post-inoculation, and two comprised of MPS1i-treated cells 10 days post-inoculation. (**F**) Top downregulated hallmark gene sets in tumors comprised of MPS1i-treated B16F10 cells, show downregulated DNA repair, cell cycle, and growth pathways, consistent with observations of slowed growth in culture and in vivo – as subsequently quantified. Gene set enrichment analysis pathways were obtained from MSigDB. Abbreviations: CL homeostasis = cholesterol homeostasis; OXPHOS = oxidative phosphorylation. (**G**) Scheme for conditioned media (secretome) treatment of BMDMs and subsequent characterization by single-cell RNA-sequencing. Media from DMSO and MPS1i-treated B16F10 cells was collected and added 1:1 with fresh media to 7 day differentiated BMDMs (with 20 ng/mL macrophage colony-stimulating factor [M-CSF]). After 24 hr, BMDMs were processed for single-cell RNA-sequencing. (**H**) (Left) Uniform manifold approximation and projection (UMAP) plots of expression profiles for all analyzed BMDMs, treated with the secretome from DMSO-treated (gray) or MPS1i-treated (red) B16F10 cells. Each circle represents an individual cell. (Right) Same UMAP plots but colors now represent cells clustered together based on similarity of global gene expression. (Inset) Composition of each cluster. (**I**) Dot plot showing proportion of cells in each cluster expressing M2-like and anti-inflammatory genes (pro-cancer gene set). Right: Heatmap scale for average gene expression and dot size reference for proportion of cells in the cluster that express a gene. Clusters 1 and 3, both of which consist of ~75% of BMDMs treated with conditioned media from DMSO-treated B16F10, generally show downregulated transcript levels for genes associated with skewing macrophages to an M1-like and pro-inflammatory phenotype (anti-cancer). Cluster 3, in particular, shows significant downregulation, whereas clusters 0 and 2 that consist of 75% of BMDMs treated with conditioned media from MPS1i-treated B16F10 show upregulated expression of many anti-cancer genes, especially cluster 2. (**J**) Dot plot showing proportion of cells in each cluster expressing M1-like and pro-inflammatory polarization-associated genes (anti-cancer gene set overall). Right: Heatmap scale for average gene expression with dot size reference for the proportion of cells in a cluster expressing a gene. Clusters 1 and 3, both of which consist of ~75% of BMDMs treated with conditioned media from DMSO-treated B16F10, generally show upregulated transcript levels for genes associated with skewing macrophages to an M2-like, anti-inflammatory, pro-cancer phenotype. Clusters 0 and 2 consist of 75% of BMDMs treated with conditioned media from MPS1i-treated B16F10 and show little-to-no expression or else downregulation of many of these genes.

The online version of this article includes the following figure supplement(s) for figure 1:

**Figure supplement 1.** Characterization of MPS1 kinase inhibition (MPS1i)-induced genome and chromosomal instability in B16F10 mouse melanoma.

a solid proliferating mass of cancer cells in non-adherent round-bottom wells (***Figure 2A***; ***Dooling et al., 2023***). We used CD47 knockout (KO) B16F10 cells, which removes the inhibitory effect of CD47 on phagocytosis, noting that KO does not perturb surface levels of Tyrp1, which is targetable for opsonization with anti-Tyrp1 (***Figure 2—figure supplement 1A***). BMDMs were added to pre-assembled tumoroids at a 3:1 ratio, and we first assessed surface protein expression of macrophage polarization markers. Consistent with our whole-tumor bulk RNA-sequencing and also single-cell RNA-sequencing of BMDM monocultures (***Figure 1E, I, and J***), BMDMs from immunotumoroids of MPS1i-treated B16F10 showed increased surface expression of M1-like markers MHCII and CD86 while showing decreased expression of M2-like markers CD163 and CD206 (***Figure 2B and C***). Although these macrophages seemed poised for anticancer activity, the cancer cells showed decreased binding of anti-Tyrp1 (***Figure 2—figure supplement 1B***) and ~20% larger size in flow cytometry (***Figure 2—figure supplement 1C***). The latter likely reflects cytokinesis defects and poly-ploidy as acute effects of CIN induction (***Chunduri and Storchová, 2019***; ***Mallin et al., 2023***). Such cancer cell changes might explain why standard 2D phagocytosis assays show BMDMs attached to rigid plastic engulf relatively few anti-Tyrp1 opsonized cancer cells pre-treated with MPS1i versus DMSO (***Figure 2—figure supplement 1D***). In such cultures, BMDMs use their cytoskeleton to attach and spread, competing with engulfment of large and poorly opsonized targets. Noting that tumors in vivo are not as rigid as plastic, our 3D immunotumoroids eliminate attachment to plastic, and large numbers of macrophages can cluster and cooperate in engulfing cancer cells in a cohesive mass (***Dooling et al., 2023***). We indeed find CIN-afflicted tumoroids are eliminated by BMDMs regardless of anti-Tyrp1 opsonization

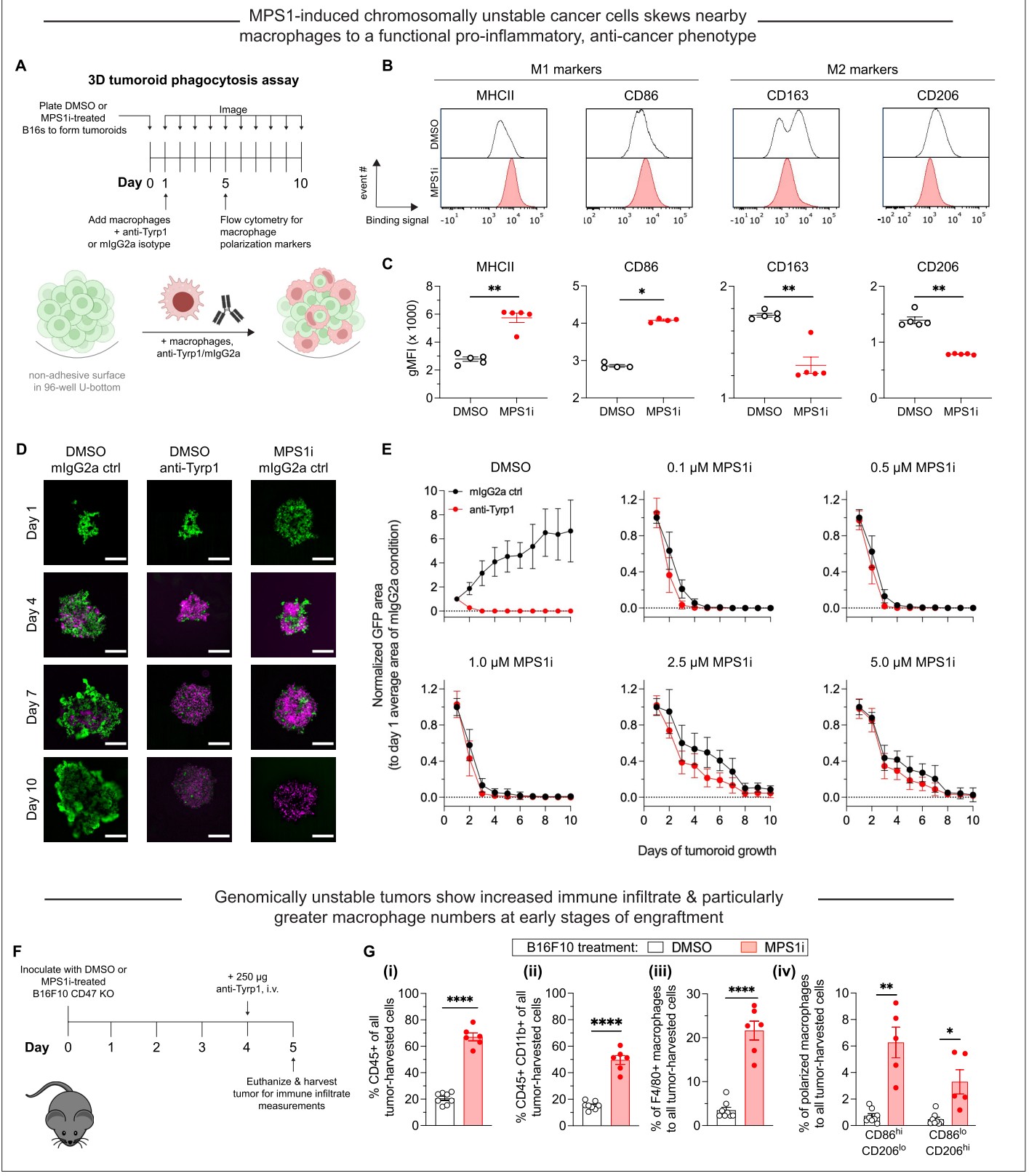

**Figure 2.** Chromosomally unstable tumoroids and tumors in early stages show increased anti-cancer macrophage polarization and activity. (**A**) Schematic for 'immuno-tumoroids'. Tumoroids are formed with B16F10 cells on non-adhesive, U-bottom-shaped wells and after pre-treating with either MPS1 kinase inhibition (MPS1i) (reversine) or DMSO. After ~24 hr, bone marrow-derived macrophages (BMDMs) with or without opsonizing anti-Tyrp1 are added to the tumoroids and imaged at the indicated timepoints. At day 5, immuno-tumoroids were collected, dissociated, and stained

*Figure 2 continued on next page*

*Figure 2 continued*

for macrophage polarization markers. (**B**) Representative flow cytometry measurements of macrophage polarization: MHCII and CD86 for M1-like (anti-tumor) markers, and CD163 and CD206 for M2-like (pro-tumor) markers. (**C**) Geometric mean fluorescence intensity of anti-tumor markers (MHCII and CD86) and pro-tumor markers (CD163 and CD206) on macrophages from immuno-tumoroid cultures of CD47 knockout (KO) B16F10 cells. Immuno-tumoroids of MPS1i-treated B16F10 CD47 KO cells show macrophages with M1-like, anti-tumor markers and reduced expression of M2-like, pro-tumoral markers. This suggests that chromosomal instability induced in cancer cells can in early stages prime a microenvironment conducive to anti-tumor macrophages, although it also reduces cancer cell proliferation. All data were collected from five independent immuno-tumoroid culture experiments (one 96-well per replicate). Statistical significance was calculated by an unpaired two-tailed t-test with Welch's correction (**p<0.01). (**D**) Representative fluorescence images of growth or repression of B16F10 CD47 KO cells (green) in immuno-tumoroids from days 1 to 10. BMDMs, shown in magenta, were added at a 3:1 ratio to initial B16F10 numbers after the day 1 images were acquired. Scale bar = 0.5 mm. (**E**) Tumoroid growth was measured by calculating the GFP+ area at the indicated timepoints (mean ± SD, n=16 total tumoroids from two independent experiments for each condition). All data were then normalized to average GFP+ area on day 1 of each drug treatment's respective mouse IgG2a isotype control condition. Overall, macrophages can clear MPS1i-treated B16F10 cells regardless of either MPS1i treatment concentration or IgG opsonization. (**F**) Timeline for immune cell infiltration analyses in tumors comprised of either MPS1i-treated or DMSO control CD47 KO B16F10 cells. Prior to tumor inoculation, cells were treated with 2.5 μM MPS1i (reversine) or the equivalent volume of DMSO vehicle control. After 24 hr, cells were washed twice with phosphate-buffered saline (PBS) and allowed to recover for an additional 48 hr. C57BL/6 mice were then subcutaneously injected with 2×10⁵ cells, and 96 hr later, mice were treated with 250 μg of anti-Tyrp1 or mouse IgG2a isotype. Mice were euthanized 24 hr later, and their tumors were excised and disaggregated for immune infiltrate analysis by flow cytometry. (**G**) Immune cell infiltrate measurements of B16F10 CD47 KO tumors comprised of MPS1i- or DMSO-treated cells. (**i**) Percentage of CD45+ (immune) cells in excised tumors shows tumors of MPS1i pre-treated cells with ~3-fold more immune cell infiltrate compared to DMSO controls. (ii) Percentage of tumor infiltrating myeloid cells in the excised tumors. CD47 KO B16F10 tumors of MPS1i pre-treated cells show ~2.5-fold more myeloid cells compared to DMSO controls. (iii) Tumor infiltrating F4/80+ macrophages (relative to total tumor cells) are ~6-fold higher for MPS1i pre-treated tumors compared to DMSO controls. (iv) M1-like macrophages (CD86$^{hi}$CD206$^{lo}$) and M2-like macrophages (CD86$^{lo}$CD206$^{hi}$) relative to the total number of macrophages, showing that MPS1i pre-treated tumors have ~6-fold more CD86$^{hi}$CD206$^{lo}$ compared to DMSO controls. MPS1i pre-treated tumors also show an increase in CD86$^{lo}$CD206$^{hi}$ macrophages, although CD86$^{hi}$CD206$^{lo}$ macrophages were twice as high. Mean ± SEM shown (n=8 mice challenged with DMSO-treated B16F10 cells, n=5 mice challenged with MPS1i-treated B16F10 cells). Statistical significance was calculated by an unpaired two-tailed t-test with Welch's correction (ns, not significant; **p<0.01; ****p<0.0001).

The online version of this article includes the following figure supplement(s) for figure 2:

**Figure supplement 1.** Characterization of MPS1 kinase inhibition (MPS1i)-induced genome and chromosomal instability in B16F10 mouse melanoma.

**Figure supplement 2.** Macrophages readily clear chromosomal instability (CIN)-afflicted tumoroids but only if CIN is accompanied by proliferation deficits.

**Figure supplement 3.** Flow cytometry gating strategy for identification and quantification of macrophage infiltrate and characterization in chromosomal instability (CIN)-afflicted and chromosomally stable B16F10 tumors.

**Figure supplement 4.** MPS1 kinase inhibition (MPS1i)-induced chromosomal instability upregulates MHC-1 class molecules on B16F10, suggesting increased antigen presentation.

(*Figure 2D and E*), whereas anti-Tyrp1 is required for clearance of DMSO control tumoroids (*Figure 2*; *Figure 2—figure supplement 2A and B*). Imaging also suggests that cancer CIN stimulates macrophages to cluster (compare day 4 in *Figure 2D*), which favors cooperative phagocytosis of tumoroids (*Dooling et al., 2023*), and occurs despite the lack of cancer cell opsonization and their larger cell size. The 3D immunotumoroid results with induced CIN are thus consistent with a more pro-phagocytic M1-type polarization (*Figures 1J and 2B, C*).

Given that our 3D immunotumoroids suggest cancer cell CIN can induce anti-cancer macrophage phenotypes to favor macrophage-mediated clearance, two other MPS1i drugs were similarly tested. Tumoroids composed of AZ3146-treated cells show that BMDMs could suppress growth at high drug concentrations (*Figure 2—figure supplement 2C – i*), whereas tumoroids composed of BAY12-17389-treated cells give results similar to reversine results (*Figure 2—figure supplement 2C – ii*). AZ3146-treated cells tended to proliferate at most drug doses, and we also verified that at least low doses of BAY 12-17389 and reversine also allowed for tumoroid growth (*Figure 2—figure supplement 2D*). The results nonetheless suggest that the anti-cancer macrophage activity observed may require CIN to be severe enough to impede proliferation. Otherwise, low-level CIN may be tolerable, eventually favoring evolution (*Vasudevan et al., 2020*), and allowing cancer cells to proliferate more rapidly than the kinetics of phagocytosis.

Lastly, prior to long-term in vivo studies, we sought to expand on some of our short-term results in our analyses of transcriptomes and tumoroid surface markers. Tumors were established in mice with either MPS1i-treated or DMSO-treated B16F10 cells (per *Figure 1D*), treated at day 4 with anti-Tyrp1, and isolated at day 5 for immune infiltrate analyses (*Figure 2F*, *Figure 2—figure supplement 3*). Flow

cytometry showed the CIN-afflicted tumors had 3-fold more CD45+ immune cells (*Figure 2G – i*), 2.5-fold more CD45+ CD11b+ (myeloid) cells (*Figure 2G – ii*), and an ~6-fold increase in F4/80-positive macrophages (*Figure 2G – iii*) relative to DMSO controls. Furthermore, we found that CIN-afflicted tumors showed approximately 6-fold more CD86$^{hi}$ CD206$^{lo}$ (M1-like, anti-cancer) macrophages compared to their DMSO counterparts. We should note that these tumors also saw an increase in CD86$^{lo}$ CD206$^{hi}$ (M2-like, pro-cancer) macrophages compared to DMSO control (*Figure 2G – iv*), but this increase is expected given that there are generally more macrophages infiltrating these tumors. Despite the increase in M2-like macrophages, these tumors made of MPS1i-treated cells still have ~2-fold more M1-like macrophages (*Figure 2G – iv*). These results suggest that although other immune cell types infiltrate tumors, which is consistent with upregulated surface expression of H2-K$^{b}$ (*Figure 2—figure supplement 4*), macrophage functional activity in vivo can again be expected to be positively affected by CIN-afflicted cancer cells.

## MPS1i-induced CIN favors tumor rejection with IgG opsonization and CD47 disruption

While previous studies show that complete CD47 ablation can favor suppression and even complete rejection of IgG-opsonized B16F10 tumors (*Andrechak et al., 2022*; *Dooling et al., 2023*; *Hayes et al., 2023*; *Kamber et al., 2021*), inter-experimental variation is high, particularly regarding complete tumor rejection and clearance. Furthermore, how CIN could influence these results is unknown. Therefore, we sought to assess if CIN in B16F10, which thus far has shown to skew macrophages toward an anti-cancer phenotype and away from pro-cancer one, can translate to improved efficacy and consistency in macrophage-oriented therapies. For these subsequent in vivo experiments, we first pre-treated B16F10 cells with either MPS1i or DMSO for 24 hr, washed the drug out, and then allowed the cells to recover for an additional 48 hr (*Figure 3A*). After the recovery period elapsed, we established tumors in mice, with either MPS1i-treated or DMSO-treated B16F10 cells. To further test for CD47-mediated effects, we used either B16F10 CD47 KO or B16F10 sgCtrl, expressing wild-type CD47 (WT) levels. Starting 4 days post-challenge, mice received either anti-Tyrp1 or mIgG2a isotype control for opsonization.

As expected, all tumors comprised of DMSO-treated B16F10 sgCtrl showed exponential growth and no survivors (*Figure 3B*), consistent with previous studies (*Dooling et al., 2023*; *Hayes et al., 2023*). These results re-confirm the inhibitory effect that CD47 has on macrophage-mediated immunity (*Ingram et al., 2017*; *Sockolosky et al., 2016*; *Willingham et al., 2012*), suppressing macrophage immune response even with anti-Tyrp1 IgG opsonization. Mice with CIN-afflicted B16F10 sgCtrl tumors ultimately showed exponential growth and no survivors as well, but they did show increased median survival. Furthermore, median survival increased even further when mice were treated with anti-Tyrp1, such that all mice were considered partial responders (survival of 20+ days, 1 week longer than median survival of tumors comprised of DMSO-treated B16F10 sgCtrl without anti-Tyrp1 treatment). All tumors comprised of DMSO-treated B16F10 CD47 KO showed exponential growth and no survivors (*Figure 3C*), regardless of anti-Tyrp1 treatment or not. However, CIN-afflicted B16F10 CD47 KO tumors showed more positive outcomes. Even without anti-Tyrp1, all challenged mice with CIN-afflicted CD47 KO tumors were either cured completely (28%) or considered partial responders. When paired with anti-Tyrp1 treatment, 97% of mice challenged with CIN-afflicted CD47 KO tumors survive.

Long-term survival results show that challenging mice with both MPS1i-treated and DMSO-treated B16F10 sgCtrl failed to yield any survivors, regardless of anti-Tyrp1 opsonization or not (*Figure 3D – i*). Similarly, we also failed to generate any survivors among mice challenged with DMSO-treated B16F10 CD47 KO cells, regardless of anti-Tyrp1 opsonization or not (*Figure 3D – ii and iii*). This again highlights a challenge in optimizing macrophage-mediated therapies: achieving consistency in long-term therapeutic outcomes. Mice challenged with MPS1i-treated B16F10 CD47 KO cells, however, were able to survive, both without anti-Tyrp1 (28% survival) and with anti-Tyrp1 (97% survival) (*Figure 3D – ii*). To determine if the degree of CIN affected survival, we also challenged mice with B16F10 CD47 KO cells treated with varying concentrations of reversine. Even at the lowest concentration that causes just 10% of cells to have micronuclei for days or less, >80% of mice survive when also treated with anti-Tyrp1 (*Figure 3—figure supplement 1*). These results show that, in physiologically relevant microenvironments, early-stage CIN can favor survival when paired with IgG opsonization and CD47 disruption. This further suggests that macrophages are key effector cells

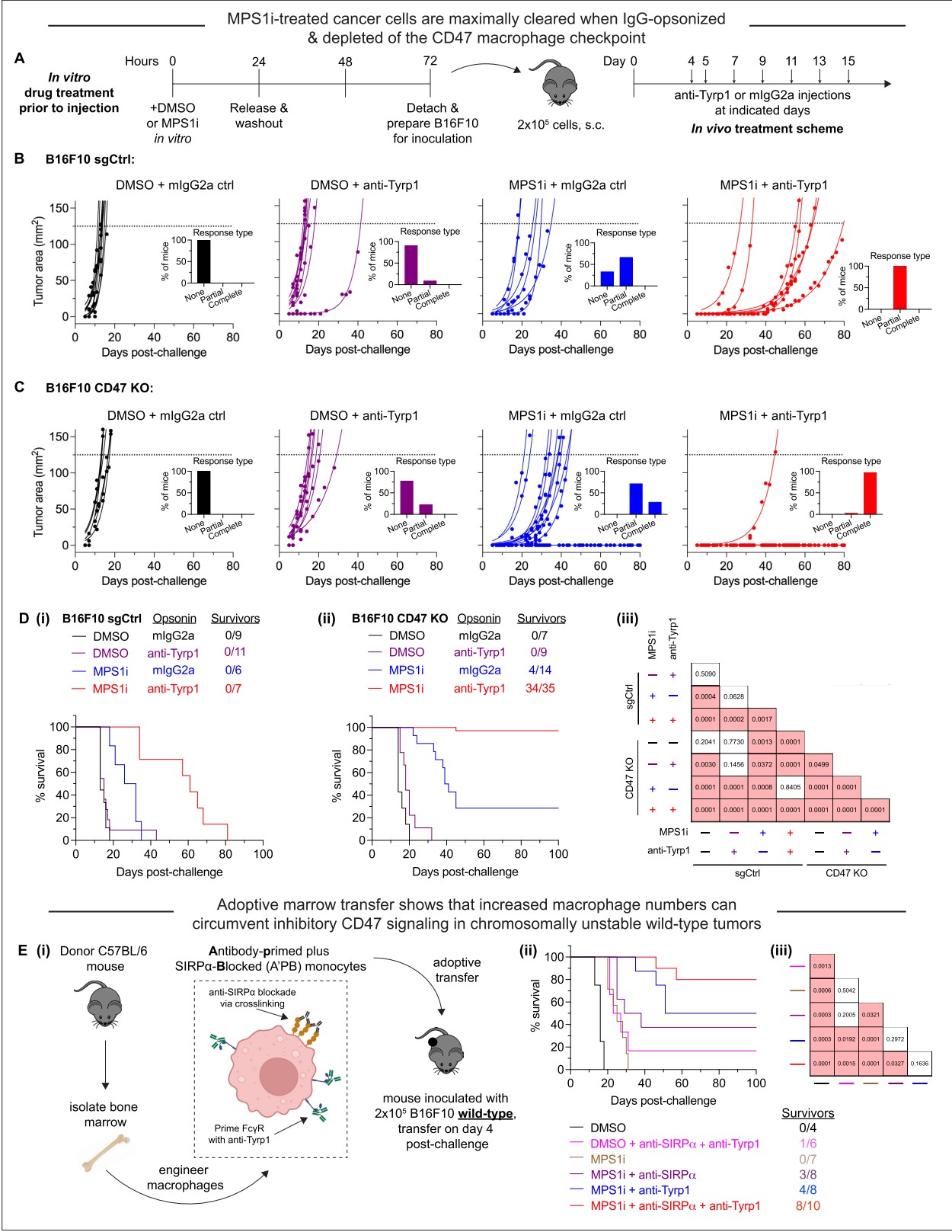

**Figure 3.** MPS1 kinase inhibition (MPS1i)-induced chromosomally unstable cancer cells are maximally cleared when IgG-opsonized and depleted of the CD47 macrophage checkpoint. (**A**) Timeline for in vitro treatment of B16F10 cells prior to injection in mice and then subsequent therapeutic treatment for tumor-challenged mice. Prior to tumor inoculation, B16F10 CD47 knockout (KO) cells were treated with 2.5 µM MPS1i (reversine) or the equivalent volume of DMSO vehicle control. Cells were treated for 24 hr, after which they were washed twice with phosphate-buffered saline (PBS) and allowed

*Figure 3 continued on next page*

*Figure 3 continued*

to recover for an additional 48 hr. After the recovery period elapsed, all mice were subcutaneously injected with $2\times10^5$ B16F10 cells. For anti-Tyrp1 and mouse IgG2a isotype control treatments, mice were treated intravenously or intraperitoneally with 250 µg with antibody on days 4, 5, 7, 9, 11, 13, and 15 after tumor challenge. (**B**) Tumor growth curve of projected tumor area versus days after tumor challenge, with B16F10 sgCtrl cells (expressing wild-type [WT] levels of CD47). Each line represents a separate tumor and is fit with an exponential growth equation: $A = A_0e^{kt}$. Experimental conditions are as follows: n = 9 mice that were challenged with DMSO-treated B16F10 sgCtrl and subsequently treated with mouse IgG2a control, n = 11 mice that were challenged with DMSO-treated B16F10 sgCtrl and subsequently treated with anti-Tyrp1, n = 6 mice that were challenged with MPS1i-treated B16F10 sgCtrl and subsequently treated with mouse IgG2a control, and n = 7 mice that were challenged with MPS1i-treated B16F10 sgCtrl and subsequently treated with anti-Tyrp1. All data were collected across three independent experiments. Inset bar graphs depict response type for each indicated tumor challenge. A partial response was defined as a mouse that survived at least 1 week (20+ days) beyond the median survival of the B16F10 sgCtrl cohort treated with mouse IgG2a isotype control (13 days). (**C**) Tumor growth curve of projected tumor area versus days after tumor challenge, with B16F10 CD47 KO cells. Each line represents a separate tumor and is fit with an exponential growth equation: $A = A_0e^{kt}$. Complete anti-tumor responses in which a tumor never grew are depicted with the same symbol as their growing counterparts and with solid lines at A = 0. Experimental conditions are as follows: n = 7 mice that were challenged with DMSO-treated B16F10 CD47 KO and subsequently treated with mouse IgG2a control, n = 9 mice that were challenged with DMSO-treated B16F10 CD47 KO and subsequently treated with anti-Tyrp1, n = 14 mice that were challenged with MPS1i-treated B16F10 CD47 KO and subsequently treated with mouse IgG2a control, and n = 35 mice that were challenged with MPS1i-treated B16F10 CD47 KO and subsequently treated with anti-Tyrp1. All data in which mice were challenged with DMSO-treated cells were collected from three independent experiments. Data for the condition in which mice were challenged with MPS1i-challenged cells and then given mouse IgG2a control were collected from four independent experiments. Data for the final condition in which mice were injected with MPS1i-treated cells and then treated with anti-Tyrp1 was collected from seven independent experiments. Inset bar graphs depict response type for each indicated tumor challenge. A partial response was defined as a mouse that survived at least 1 week (20+ days) beyond the median survival of the B16F10 sgCtrl cohort treated with mouse IgG2a isotype control (13 days). (**D**) Survival curves of mice up to 100 days after the tumor challenges in (**B**) and (**C**). (**i**) Survival curves for mice challenged with B16F10 sgCtrl. (ii) Survival curves for mice challenged with B16F10 CD47 KO. (iii) Triangular matrix depicting p-values between the different tested in vivo conditions. Statistical significance was determined by the Log-rank (Mantel-Cox) test. (**E**) (**i**) Schematic depicting the different engineering anti-tumor macrophage strategies that can be used for validating macrophages' role in clearing chromosomal instability (CIN)-afflicted B16F10 cells. Fresh bone marrow was isolated from the tibia of donor C57BL/6 mice and was then incubated with anti-SIRPα clone P84 (18 µg/mL) or anti-Tyrp1 (1 µg/mL) only, both together added subsequently, or neither. Donor marrow ($2\times10^7$ cells) is then injected intravenously into C57BL/6 harboring B16F10 WT tumors on day 4 post-challenge. Additional anti-Tyrp1 injections, when necessary, are done on days 5, 7, 9, 11, 13, and 15 post-tumor challenge. (ii) Survival curves of mice up to 100 days after the B16F10 WT tumor challenge. All mice were initially challenged with $2\times10^5$ B16F10 WT cells. (iii) Triangular matrix depicting p-values between the different tested in vivo conditions. Statistical significance was determined by the Log-rank (Mantel-Cox) test.

The online version of this article includes the following figure supplement(s) for figure 3:

**Figure supplement 1.** MPS1 kinase inhibition (MPS1i)-induced chromosomal instability (CIN) favors clearance when paired with CD47 knockout (KO) and IgG opsonization, regardless of the degree of CIN.

in achieving survival against CIN-afflicted tumors, since 97% survival was achieved under conditions of maximal phagocytosis.

Although no mice challenged with CIN-afflicted B16F10 sgCtrl survived, MPS1i increase median survival significantly when paired with anti-Tyrp1 opsonization (*Figure 3D – i*). This suggests that macrophages still display some anti-cancer activity, despite the inhibitory effects of CD47, and are important effector cells in final therapeutic outcome. To better support the hypothesis that macrophages are indeed key effector cells in rejecting CIN-afflicted tumors, we established tumors comprised of WT B16F10 in mice. Although WT tumors are generally unaffected by anti-CD47 and anti-Tyrp1 (*Dooling et al., 2023*; *Hayes et al., 2023*), we hypothesized that we could eliminate CIN-afflicted WT B16F10 tumors by providing adoptive transfer of marrow to increase macrophage numbers to compensate for the CD47-mediated inhibition of endogenous macrophages. Furthermore, we engineered marrow by priming Fcγ receptors with anti-Tyrp1, initially inhibiting CD47-SIRPα interaction via anti-SIRPα antibody blockade or combining both (*Figure 3E – i*). Mice challenged with CIN-afflicted WT tumors and treated with unprimed marrow all succumbed, surviving longer than controls (*Figure 3E – ii and iii*) and similar to mice that did not receive marrow (blue curve in *Figure 3D – i*). Chromosomally stable (DMSO) WT tumors treated with marrow primed with both anti-SIRPα and anti-Tyrp1 were statistically the same, despite one survivor (17%). Increased macrophage numbers thus provide little to no benefit on their own. However, 37% of mice challenged with CIN-afflicted WT tumors survived when treated with marrow that was initially engineered with anti-SIRPα, and 50% survived with marrow primed with anti-Tyrp1. Anti-Tyrp1 without adoptive transfer of marrow gave no long-term survivors (red curve in *Figure 3D – i*), which highlights a key role for infusions of marrow myeloid cells. Intriguingly, 80% of mice survived with anti-SIRPα-blocked marrow myeloid cells primed with anti-Tyrp1 (*Figure 3E – ii and iii*). These results together support three conclusions for CIN-driven tumors: (1) increased survival

attributed to anti-SIRPα supports the idea that CD47 modulates therapeutic outcome even during CIN; (2) increased survival attributed to anti-Tyrp1 highlights the importance of IgG opsonization; (3) maximal survival with the combination emphasizes that macrophages – which are skewed by CIN – play a key role in achieving near-complete rejection of tumor.

## Clearance of CIN-afflicted tumors promotes de novo pro-phagocytic and anti-cancer IgG

Macrophages and related phagocytic myeloid cells constitute a first line of innate immune defense against pathogens and disease, but some can also initiate acquired immunity. Two key branches of such immunity are humoral immunity mediated by macromolecules such as antibodies and cell-mediated immunity involving T cells, for example. We hypothesized that mice that survived challenges with CIN-afflicted tumors would show signs of acquired immunity due to their high survival rate, and we focused on IgG because of our many functional assays.

We collected convalescent serum from survivors to quantify de novo anti-cancer IgG antibodies that possibly resulted from successful rejection and clearance of CIN-afflicted tumors (*Figure 4A*). Sera were collected and then subsequently used in antibody binding testing and western blotting to assess emergence of anti-B16F10 antibodies. We first quantified IgG2a and IgG2b titers in convalescent sera, both of which have been previously found to engage mouse macrophage Fcγ receptors (*Bruhns, 2012*; *Nimmerjahn et al., 2010*) that are typically required for macrophage-mediated phagocytosis. B16F10 cells, either Tyrp1-expresing or Tyrp1 KO, were incubated with sera (convalescent from survivors or naïve from unchallenged mice) and then counterstained with conjugated antibodies against IgG2a/c and IgG2b (*Figure 4B – i and ii*). All mice that survived challenges from CIN-afflicted tumors yielded sera that showed significantly large increases in IgG2a/c binding against both Tyrp1-positive and Tyrp1 KO B16F10 cells. We similarly saw a statistically significant increases in IgG2b binding against both Tyrp1-positive and Tyrp1 KO B16F10 cells using the same sera, although the increases in binding were more variable. The increases in binding observed against Tyrp1 KO cells also suggest that IgG antibodies target a repertoire of antigens beyond Tyrp1, which we further qualitatively confirmed via western blotting (*Figure 4B – iii*). Convalescent sera were used to immunoblot against B16F10 lysates, revealing many bands at multiple molecular weights and more bands than when immunoblotting with naïve sera, supporting our hypothesis of antigen broadening beyond Tyrp1. Lastly, we also tested IgG2a/c and IgG2b titers from additional in vivo experiments: survivors from both titrated CIN-afflicted tumors from *Figure 3—figure supplement 1* and adoptive marrow transfers in *Figure 3E*. Similarly, convalescent sera from all these mice show increases in IgG2a/c and IgG2b in both Tyrp1-expressing and Tyrp1 KO B16F10 cells (*Figure 4—figure supplement 1*). These results confirm that regardless of the method used to exploit CIN, induction of anti-cancer IgG can be expected.

To test whether these de novo serum antibodies functionally promote macrophage-mediated phagocytosis, we performed conventional 2D phagocytosis assays in which cancer cell suspensions were opsonized with sera (or anti-Tyrp1 or mouse IgG2a isotype as controls) (*Figure 4C – i*). Under conditions of CD47 KO, we see that nearly all unpurified convalescent sera increased phagocytosis relative to naïve serum and mIgG2a isotype control. Furthermore, this increase in phagocytosis is identical to that provided by anti-Tyrp1 (both ~5-fold higher than baseline). We also find that sera continue to promote phagocytosis even in B16F10 CD47/Tyrp1 double KO cells (~3-fold higher than baseline). Convalescent sera are still able to increase phagocytosis against double KO cells, whereas anti-Tyrp1 expectedly does not drive phagocytosis due to lack of antigen. This again supports the hypothesis of acquired immunity with de novo IgG antibodies that target B16F10 antigens beyond Tyrp1.

Upon confirming the functional effect of de novo IgG in the convalescent sera, we then wondered if convalescent serum IgG would be able to suppress tumoroid growth, given that this model better captures both the biophysical microenvironment and proliferative capacity of tumors (*Dooling et al., 2023*). Indeed, we found that convalescent serum IgG added simultaneously with macrophages to B16F10 CD47 KO tumoroids led to either tumoroid elimination or significantly suppressed growth (*Figure 4C – ii*), although the efficacy was less potent than anti-Tyrp1. This suggests that perhaps the polyclonal de novo IgGs here still lack the specificity benefits that accompany a monoclonal antibody such as anti-Tyrp1. Nonetheless, these results demonstrate induction of a generally potent anti-cancer antibody response to CIN-afflicted B16F10 in a CD47 KO context. Importantly, comparing these sera

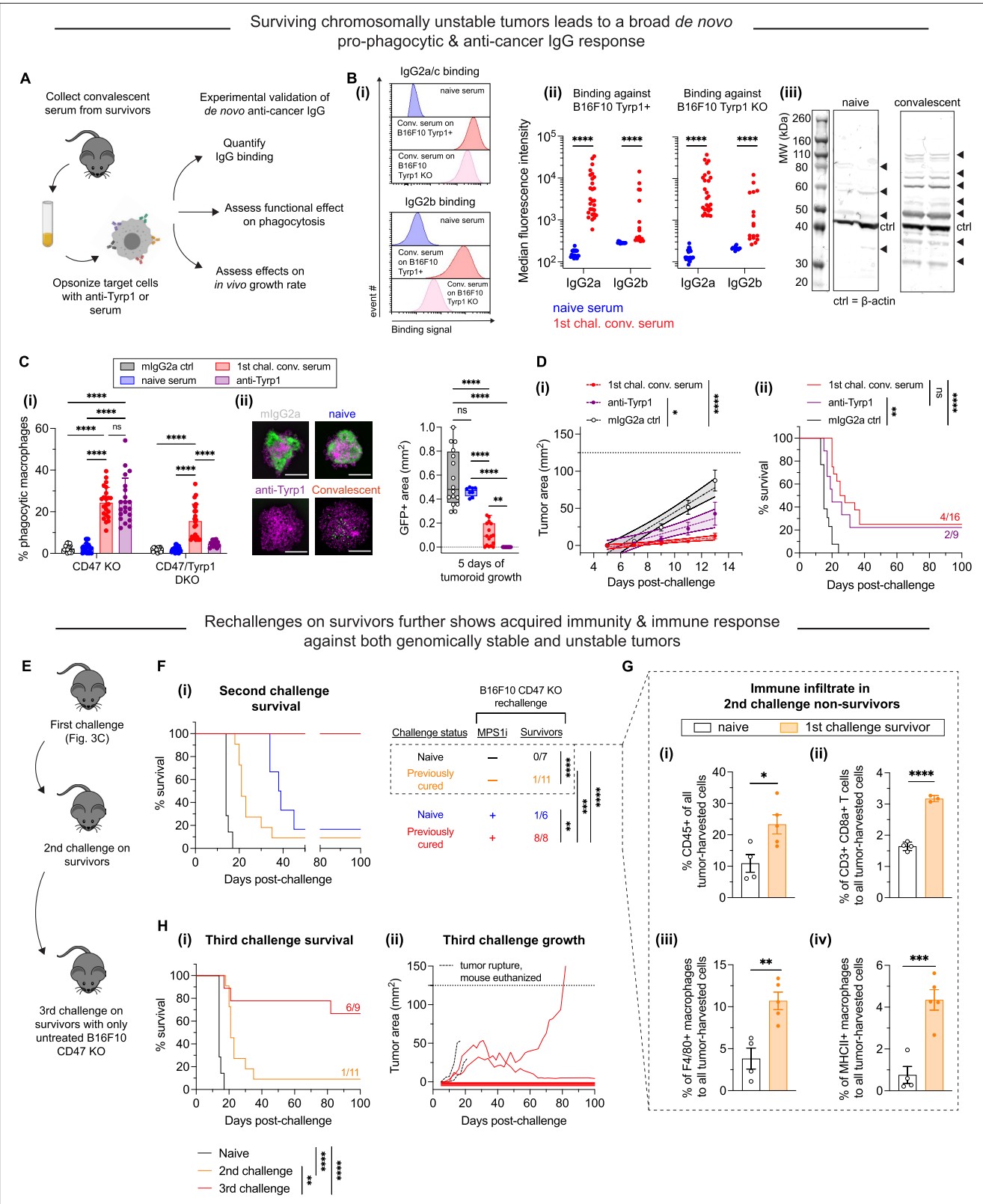

**Figure 4.** MPS1 kinase inhibition (MPS1i)-induced chromosomal instability favors induction of pro-phagocytic de novo IgG and can lead to durable acquired immunity. (**A**) Schematic for sera collection from surviving mice from *Figure 3C and D* and follow-up experiments to characterize any de novo anti-cancer IgG antibodies and their functionality in vitro and in vivo. Serum from all mice was collected at least 100 days after initial tumor challenge. (**B**) (**i**) Representative flow cytometry histograms showing convalescent sera from survivors in *Figure 3C and D* contain IgG2a/c (top) and IgG2b (bottom)

*Figure 4 continued*

that bind both wild-type (WT) and Tyrp1 knockout (KO) B16F10 cells. (ii) Median fluorescence intensity quantification of IgG2a/c and IgG2b binding from sera of surviving mice, showing a significant increase in both. Binding to Tyrp1 KO cells indicates recognition of other antigens. Statistical significance was calculated by an unpaired two-sample Kolmogorov-Smirnov test (****p<0.0001). For IgG2a/c quantification: for binding against Tyrp1+ cells, n = 23 distinct naïve serum samples and n = 28 distinct convalescent serum samples from surviving mice; for binding against Tyrp1 KO cells, n = 23 distinct naïve serum samples and n = 27 distinct convalescent serum samples. For IgG2b quantification: for binding against Tyrp1+ cells, n = 14 distinct naïve serum samples and n = 16 distinct convalescent serum samples from surviving mice; for binding against Tyrp1 KO cells, n = 13 distinct naïve serum samples and n = 17 distinct convalescent serum samples. (iii) Western blot of B16F10 lysate with either naïve sera or first challenge survivor sera as primary probe followed by anti-mouse IgG [H+L] secondary staining. Numerous bands appear when immunoblotting with convalescent survivor sera compared to naïve sera, confirming multiple neo-antigens and suggesting acquired immunity beyond Tyrp1. (**C**) (**i**) Phagocytosis of serum-opsonized CD47 KO or CD47/Tyrp1 double KO B16F10 cells by bone marrow-derived macrophages (BMDMs) on 2D tissue culture plastic. Additionally, B16F10 cells opsonized with either anti-Tyrp1 or mouse IgG2a were included as controls for comparisons. Serum IgG derived from survivors has both opsonization and pro-phagocytic functional ability against B16F10. Furthermore, convalescent sera IgG from survivors is still able to drive engulfment of Tyrp1 KO cells, further suggesting targeting of antigens beyond Tyrp1. Statistical significance was calculated by two-way ANOVA and Tukey's multiple comparison test (mean ± SD, n = 21–23 distinct sera samples collected from survivors for B16F10 CD47 KO phagocytosis per condition and n = 14–23 for B16F10 CD47/Tyrp1 double KO phagocytosis per condition). (ii) Convalescent sera from first challenge survivors can repress growth of B16F10 CD47 KO immuno-tumoroids (with macrophages). Tumoroid growth was measured by calculating the GFP+ area at the indicated timepoints (mean ± SD, n=16 total tumoroids from two independent experiments for each condition, except n=8 for opsonization with sera from naïve mice). Statistical significance was calculated by Brown-Forsythe and Welch ANOVA tests with Dunnett's T3 corrections for multiple comparisons (ns, not significant; **p<0.01; ****p<0.0001). Scale bars = 0.5 mm. (**D**) B16F10 CD47 KO cells were pre-opsonized with either convalescent sera from first challenge survivors, anti-Tyrp1 or mouse IgG2a isotype control. All mice were subcutaneously injected with 2×10⁵ pre-opsonized B16F10 CD47 KO cells. (**i**) Tumor growth at early timepoints where growth is still linear shows suppression of tumors comprised of B16F10 CD47 KO cells pre-opsonized with convalescent sera and anti-Tyrp1, compared to mouse IgG2a isotype controls. Mean ± SEM for all timepoints, with n=16 mice with tumors pre-opsonized with convalescent sera (each from a distinct survivor), n=9 mice with tumors pre-opsonized with anti-Tyrp1, and n=14 mice with tumors pre-opsonized with mouse IgG2a isotype control. Statistical significance was calculated by ordinary one-way ANOVA and Tukey's multiple comparison test at days 9, 11, and 13 (*p<0.05; ****p<0.0001). Significance represented in plot legend is representative of all three timepoints. (ii) Survival curves up to 100 days of mice from (**D – i**) with pre-opsonized tumors. Both convalescent sera and anti-Tyrp1 provide similar survival benefits, suggesting potent de novo IgG opsonization and anti-cancer function. Statistical significance was determined by the Log-rank (Mantel-Cox) test (ns, not significant; **p<0.01; ****p<0.0001). (**E**) Schematic of tumor challenges to assess acquired immunity. Survivors from the first challenge (*Figure 3C and D*) were again challenged with either MPS1i-treated or DMSO B16F10 CD47 KO cells. Survivors from this second tumor challenge were once again challenged, this time with untreated B16F10 CD47 KO. (**F**) Survival curves of survivors from *Figure 3C and D* for a second tumor challenge experiment. B16F10 CD47 KO cells were pre-treated for 24 hr with 2.5 µM MPS1i (reversine) or the equivalent volume of DMSO vehicle control. Cells were then washed twice with phosphate-buffered saline (PBS) and allowed to recover for 48 hr. Mice were then subcutaneously injected with 2×10⁵ B16F10 CD47 KO cells. n=7 age-matched naïve mice (never tumor-challenged) injected with DMSO-treated B16F10 CD47 KO cells, n=11 surviving mice (from *Figure 3C and D*) injected DMSO-treated B16F10 CD47 KO cells, n=6 age-matched naïve mice injected with MPS1i-treated B16F10 CD47 KO cells, and n=8 surviving mice (from *Figure 3C and D*) injected with MPS1i-treated B16F10 CD47 KO cells. Previous survivors challenged with DMSO-treated B16F10 CD47 KO cells show increased median survival (21 days) compared to their naïve counterpart (14 days). All previous survivors that were again challenged with MPS1i-treated B16F10 CD47 KO cells survive. All mice challenged were from three independent experiments. Statistical significance was determined by the Log-rank (Mantel-Cox) test (**p<0.01; ***p<0.001; ****p<0.0001). (**G**) Non-survivors from the second tumor challenge in (**F**) were euthanized after tumor size was >150 mm², and their tumors were excised and disaggregated for immune infiltrate analysis by flow cytometry. (**i**) Quantification of CD45+ (immune) cells in the excised tumors shows first challenge survivors with ~2.5-fold increased tumor immune cell infiltrate despite terminal burden. n = 4 mice for age-matched naïve control, n = 5 mice that survived the first tumor challenge. (ii) Quantification of tumor infiltrating CD8a+ cytotoxic T cell relative to the total number of tumor cells. First challenge survivors show ~2-fold increase in CD8a+ T cells. n = 4 mice for age-matched naïve control, n = 3 mice that survived the first tumor challenge. (iii) Quantification of tumor infiltrating F4/80+ macrophages relative to the total number of tumor cells. First challenge survivors show ~3-fold increase in macrophages. n = 4 mice for age-matched naïve control, n = 5 mice that survived the first tumor challenge. (iv) Quantification of MHCII+ tumor infiltrating F4/80+ macrophages relative to the total number of F4/80 macrophages. First challenge survivors show ~3-fold increase in MHCII+ macrophages. n = 4 mice for age-matched naïve control, n = 5 mice that survived the first tumor challenge. For all experiments, mean ± SEM shown, and statistical significance was calculated by an unpaired two-tailed t-test with Welch's correction (*p<0.05; **p<0.01; ***p<0.001; ****p<0.0001). (**H**) (**i**) Third tumor challenge survival curves of long-term survivors from (**G**). All mice were challenged with 2×10⁵ B16F10 CD47 KO cells (n = 9 mice from three independent experiments), delivered subcutaneously. For benchmarking and statistical comparison, survival curves from *Figure 3F* for naïve mice (n = 7 mice from three independent experiments) and second challenge (n = 11 mice from three independent experiments) are included. Long-term survivors challenged a third time show ~70% survival without any additional therapeutic modality, suggesting significantly improved acquired immune response. Statistical significance was determined by the Log-rank (Mantel-Cox) test (**p<0.01; ***p<0.001; ****p<0.0001). (ii) Individual tumor growth curves for third challenge in long-term survivors (n = 9 mice from three independent experiments) shown in (**H – i**). In total, four mice developed tumors, two of which had to be euthanized prematurely due to tumor rupture despite not reaching a terminal burden of 125 mm². These mice are included in the survival analysis. The two remaining mice show significantly slower tumor growth than naïve mice challenged with regular B16F10 CD47 KO (median survival of 14 days) and can be considered partial responders.

The online version of this article includes the following figure supplement(s) for figure 4:

*Figure 4 continued on next page*

Figure 4 continued

**Figure supplement 1.** Survivors challenged with chromosomal instability (CIN)-afflicted tumors generate anti-cancer IgG, regardless of the degree of CIN.

**Figure supplement 2.** Flow cytometry gating strategy for identification and quantification of immune infiltrate and characterization in re-challenge experiments.

**Figure supplement 3.** Growth of chromosomal instability (CIN)-afflicted wild-type (WT) tumors in T- and B-cell deficient mice and T- and B-cell replete mice.

results for CIN-afflicted tumors to our recent studies of the same tumor model without CIN (*Dooling et al., 2023*; *Hayes et al., 2023*), we find similar levels of IgG induction (e.g. ~100-fold above naïve on average for IgG2a/c), similar increases in phagocytosis by sera opsonization (e.g. equivalent to anti-Tyrp1), and similar levels of suppressed tumoroid growth – including the variability.

We then proceeded to test the function of convalescent serum in vivo by opsonizing B16F10 CD47 KO cells just prior to subcutaneous implantation in naïve mice. For comparison, we also opsonized B16F10 CD47 KO cells with either anti-Tyrp1 (positive control) or mIgG2a control (negative control). We found that both convalescent sera and anti-Tyrp1 suppressed tumor growth by days 11 and 13 relative to mIgG2a control (*Figure 4D – i*). Interestingly, we found that convalescent sera showed a trend of suppressing growth more than anti-Tyrp1, although this was not statistically significant. We continued to monitor all mice for long-term survival, and we found that this pre-opsonization with both convalescent sera and anti-Tyrp1 eliminated tumors in challenged mice with near identical cure rates (25% and 22%, respectively) (*Figure 4D – ii*). Altogether, we found that the convalescent sera from mice originally challenged with chromosomally unstable tumors has potent anti-cancer effects in vitro and in vivo.

## Acquired immunity suppresses growth of chromosomally stable tumors and becomes more effective with ongoing challenges of CIN-afflicted tumors

The anti-cancer IgG antibody development in survivors led us to further hypothesize that we should see improved median survival and/or survival rate if surviving mice were re-challenged. We therefore challenged surviving mice with a second injection of either DMSO-treated or MPSi-treated B16F10 CD47 KO cells (*Figure 4E*). If additional survivors resulted from this experiment, we also intended to undertake a third challenge, akin to a prime-and-boost strategy for anti-cancer vaccination. It should be highlighted that starting from this second challenge, no mice received anti-Tyrp1. This was done to maximally challenge acquired immunity and to better simulate the possibility of recurrence post-therapy. Age-matched naïve mice receiving their first challenge responded similarly to the younger cohorts (*Figure 3B*). Of the previously cured mice, only a single mouse (of 11 total) survived a challenge with DMSO-treated B16F10 CD47 KO cells (*Figure 4F*). However, median survival increased (21 days) compared to their naïve counterparts (14 days), supporting the initial hypothesis of prolonged survival and consistent not only with past results indicating major benefits of a prime-and-boost approach with anti-Tyrp1 (*Dooling et al., 2023*) but also with the noted similarities in induced IgG levels. Survivors that were re-challenged again with MPS1i-treated cells, however, showed 100% survival, even in the absence of anti-Tyrp1 (*Figure 4F*). Age-matched naïve mice receiving their first challenge of MPS1i-treated cells responded relatively similarly (a single survivor out of 6 total, 17% survival) to the younger cohort (28% survival). This complete success rate against a second challenge of CIN-afflicted B16F10 CD47 KO further supports an acquired immune response, at least against ongoing chromosomally unstable cells.

Mice that failed to reject re-challenge tumors comprised of DMSO-treated B16F10 CD47 KO in *Figure 4F* were euthanized and had their tumors harvested to measure their immune cell infiltrate by flow cytometry (*Figure 4—figure supplement 2*). Although these mice did not survive, we found that these previous survivors showed roughly a 2-fold increase in the number of immune cells in their tumors (*Figure 4G – i*), a 2-fold increase in the number of CD3+ CD8+ T cells (*Figure 4G – ii*), and a near 3-fold increase in the number of F4/80+ macrophages (*Figure 4G – iii*) compared to their naïve counterparts. We further found that of the F4/80+ macrophage infiltrate, previous survivors had roughly four times as many MHCII-high macrophages (M1-like, anti-cancer). These immune infiltrate

analyses suggest that although the acquired immune response in these previous survivors was still not potent enough to clear chromosomally stable and normally proliferating B16F10 CD47 KO tumors, it did enhance cell-mediated immunity.

Lastly, we performed a third tumor challenge on second challenge survivors with untreated B16F10 CD47 KO cells (chromosomally stable and regular proliferating). Mice were left untreated post-challenge (no anti-Tyrp1). ~56% of these mice completely resisted tumor growth (*Figure 4H – i*). Of the four mice that developed tumors, two had to be euthanized prematurely due to tumor rupture but were indeed showing signs of growth suppression (*Figure 4H – ii*). The last two mice consisted of two long-term partial responders: one whose tumor did not reach terminal burden until 82 days post-challenge and another who experienced stable tumor regression with almost no regrowth. Overall, the 56% survival rate in the third challenge, in the absence of anti-Tyrp1, and the partial responses observed in mice that grew tumors suggest a durable immunological response that results from encountering and clearing CIN tumors, at least in the context of CD47 disruption.

Macrophages seem to be the key *initiating*-effector cells, based in part on the following findings. First, macrophages and related myeloid cells with both SIRPα blockade and FcR-engaging, tumor-targeting IgG maximize survival of mice with WT B16+Rev tumors (*Figure 3E*) – noting that macrophages express SIRPα and FcRs, but most or all T cells do not. Despite the clear benefits of adding macrophages, to further assess whether T and B cells are key initiating-effector cells, new experiments were done with mice lacking T and B cells. We compared the growth delay of MPS1i versus DMSO treatments in these mice to the delay in fully immunocompetent mice with T and B cells – with all studies done at the same time. We found that slower growth with Rev relative to DMSO was similar in mice without T and B cells when compared to immunocompetent C57 mice (*Figure 4—figure supplement 3*). We conclude therefore that T and B cells are not key initiating-effector cells. At later times, B cells are likely effector cells at least in terms of making anti-tumor IgG, and T cells in tumor re-challenges are also increased in number (*Figure 4G – ii*). We further note that in our earlier collaborative study (*Harding et al., 2017*) WT B16 cells were pre-treated by genome-damaging irradiation before engraftment in C57 mice, and these cells grew minimally – similar to MPS1i treatment – while untreated WT B16 cells grew normally at a contralateral site in the same mouse. Such results indicate that T and B cells in C57BL/6 mice are not sufficiently stimulated by genome-damaged B16 cells to generically impact the growth of undamaged B16 cells.

## Discussion

Macrophage-directed immunotherapies against solid tumors have the potential for maximal efficacy when at least three elements are combined: large numbers of macrophages for cooperativity (in clusters), IgG opsonization that activates Fc receptors and stimulates macrophage-mediated phagocytosis, and disruption of the CD47 macrophage immune checkpoint (*Dooling et al., 2023*). Screens might add to this as might discoveries in aneuploidy, but even for well-established factors properly applied, complete tumor rejection and clearance is not guaranteed and varies greatly across in vivo mouse studies (*Andrechak et al., 2022*; *Dooling et al., 2023*; *Hayes et al., 2023*; *Cohen-Sharir et al., 2021*; *Kamber et al., 2021*). Here, we show a 97% survival rate of immunocompetent mice challenged with CIN-afflicted, syngeneic B16F10 tumors when maximizing macrophage-mediated activity. The result is notable given that B16F10 tumors are poorly immunogenic, do not respond to either anti-CD47 or anti-PD-1/PD-L1 monotherapies, and show modest and variable cure rates (~20–40%; *Dooling et al., 2023*; *Hayes et al., 2023*) even when macrophages have been made maximally phagocytic according to notions above. We should note here that our whole-tumor RNA-sequencing data (*Figure 1E*) shows that expression of PD-1 (*Pdcd1*) follows no consistent trend upon MPS1i treatment, and that *Pdcd1* was not detected in our single-cell RNA-sequencing data for macrophage cultures (*Figure 1G*) – motivating further study.

These results suggest that CIN in early stages generates anti-cancer vulnerabilities that favor macrophage-mediated immune response, contingent on conditions for maximal phagocytosis. Immunocompetent mice consistently survive these challenges at high survival rates. These survivors also develop de novo anti-cancer IgG, similar to previous studies (*Dooling et al., 2023*; *Hayes et al., 2023*), that are pro-phagocytic, multi-epitope, and efficacious in vivo. The emergence of these IgGs could be synergistic with clinically relevant CD47 blockade treatments for solid tumors and help address concerns regarding resistance due to antigen loss (*Jalil et al., 2020*). Mice that

are re-challenged with chromosomally stable CD47 KO tumors (and without exogenous anti-Tyrp1 opsonization) show increased median survival and increased immune infiltrate, further supporting the hypothesis of newly generated anti-cancer acquired immunity. More interestingly, though, we see that all mice re-challenged with CIN-afflicted CD47 KO tumors survive, even in the absence of anti-Tyrp1 opsonization. A third challenge of these two-time survivors with chromosomally stable CD47 KO tumors shows >50% mice survive and improved median survival for non-survivors. These results elucidate at least two advantages that CIN can provide to better therapeutic outcomes. First, early-stage CIN facilitates survival while generating potent de novo IgGs that can drive positive phagocytic feedback to minimize recurrence. Second, ongoing vulnerability-inducing CIN in tumor cells can both strengthen cell-mediated acquired immunity and create an antigen reservoir for the maintenance of long-term humoral immunity.

The effects of CIN and aneuploidy in macrophages certainly require further investigation. We did publish initial results showing M1-like polarization of BMDMs with IFNγ priming is sufficient to suppress growth of B16 tumoroids with anti-Tyrp1 opsonization more rapidly than unpolarized/unprimed macrophages and much more rapidly than M2-like polarization of BMDMs with IL4 (Extended Data Fig.5a in *Dooling et al., 2023*); hence, anti-cancer polarization contributes in this assay. While the secretome from MPS1i-treated cancer cells has been found to trigger expression of *Arg1* and *Il6* (*Xian et al., 2021*), both of which are pro-cancer M2-like macrophage markers (*Fernando et al., 2014*; *Mujal et al., 2022*), our findings suggest that polarization is much more complex. Here, whole-tumor bulk RNA-sequencing hints at CIN-afflicted tumors having a macrophage population that is both less anti-inflammatory and M2-like. Single-cell RNA-sequencing of BMDMs treated with secretome from CIN-afflicted cells further suggests that CIN induces a microenvironment that can push macrophages to a pro-inflammatory, anti-cancer phenotype while minimizing polarization to a pro-cancer phenotype. Our transcriptomics analyses also align more with deeper investigation that suggest additional markers are required for macrophage polarization distinction (*Jablonski et al., 2015*). We further confirm these findings by observing increased surface protein expression of anti-cancer M1-like macrophages in in vitro 3D tumoroid co-cultures with CIN-afflicted cells and in vivo immune infiltrate experiments. Functional tests are also crucial: BMDMs show enhanced clearance of CIN-afflicted cells in 3D tumoroid phagocytosis assays. Additionally, the aforementioned study (*Xian et al., 2021*) only found this trend in SKOV3 cells, whereas their aneuploid fused B16 cells show decreased *Arg1* expression, suggesting possible cell-intrinsic complications. Marker-based meta-analyses of aneuploid tumor data from TCGA also suggested reduced anti-cancer macrophage activity (*Davoli et al., 2017*; *Xian et al., 2021*), but TCGA studies are likely limited to late timepoints when cancer cells have overcome CIN- and aneuploidy-associated stresses and selected for those aneuploidies that drive tumor progression in the diagnosed patients. More recently, for some solid tumors, aneuploidy with high mutational burdens reportedly shows increased survival in immunotherapy patients (*Spurr et al., 2022a*; *Spurr et al., 2022b*). Assuming that progress can be made on clinically safe targeting of the macrophage checkpoint, to overcome some current issues, the results here also suggest interesting effects and possible benefits when combining cancer CIN with sufficient myeloid cells, disruption of CD47-SIRPα, and tumor opsonization.

# Materials and methods
## Cell lines and culture

B16F10 cells (CRL-6475) were obtained from American Type Culture Collection (ATCC). Cells were cultured at 37°C and 5% $CO_2$ in either RPMI-1640 (Gibco 11835-030) or Dulbecco's Modified Eagle Medium (DMEM, Gibco 10569-010) supplemented with 10% fetal bovine serum (FBS, Sigma F2442), 100 U/ mL penicillin, and 100 µg/mL streptomycin (1% P/S, Gibco 15140-122). Cells were passaged every 2–3 days when a confluency of ~80% was reached. For trypsinization, cells were washed once with Dulbecco's phosphate-buffered saline (PBS, Gibco 14190-136) and then detached with 0.05% Trypsin (Gibco 25300-054) for 5 min at 37°C and 5% $CO_2$. Trypsin was quenched with an equal volume of complete culture media. B16F10 cells were maintained in passage in RPMI-1640 but switched to DMEM at least 3 days prior to in vivo subcutaneous injections.

## Cell line authentication

B16F10 melanoma cells were derived decades ago from a C57BL/6 mouse and are not on the list of commonly misidentified cell lines maintained by the International Cell Line Authentication Committee. Nonetheless, per good practice and eLife instructions, cell line authentication of non-human cell lines requires verification of cell line identity, including statements of (1) source, (2) authentication method(s), and ultimately (3) confirmation that the identity has been authenticated. (1) Source: B16F10 cells (CRL-6475) were obtained from ATCC. (2) Authentication: We visualize B16F10 melanization in culture and in tumors as expected of this pigmented melanoma. We also show rapid growth of subcutaneous tumors in C57BL/6 mice is consistent with historical data from other labs (*Dooling et al., 2023*). We analyzed mouse-specific sequencing data and marker expression that also align with established profiles of the B16F10 cell line (e.g. Tyrp1, Cd47, Sirpα) as presented here and in our recent studies (*Dooling et al., 2023*; *Hayes et al., 2020*). (3) Confirmation: Based on the above and further tests showing the B16F10 cell line is negative for mycoplasma contamination (U Penn Cell Center testing in triplicate, relative to positive and negative controls), we confirm that the cell line identity has been authenticated as that of B16F10.

## Antibodies

Antibodies used for in vivo treatment and blocking and for in vitro phagocytosis are as follows: anti-mouse/human Tyrp1 clone TA99 (BioXCell BE0151), mouse IgG2a isotype control clone C1.18.4 (BioXCell BE0085), and Ultra-LEAF anti-mouse CD172a (SIRPα) clone P84 (BioLegend 144036). Low-endotoxin and preservative-free antibody preparations were used for in vivo treatments and in vitro phagocytosis experiments. For primary antibody staining of surface proteins via flow cytometry, the following were used: anti-mouse CD47 clone MIAP301 (BioXCell BE0270) and anti-mouse/human Tyrp1 clone TA99. Secondary antibodies used for flow cytometry are as follows: Alexa Fluor 647 donkey anti-mouse IgG (Thermo Fisher A-31571) and Alexa Fluor 647 goat anti-rat IgG (Thermo Fisher A-21247). All secondary antibody concentrations used following the manufacturer's recommendations. Please see Key resources table for further details.

For immune infiltrate analysis, the following BioLegend antibodies were used: Brilliant Violet 650 anti-mouse CD45 clone 30-F11 (103151), Brilliant Violet 785 anti-mouse CD45 clone 30-F11 (103149), APC/Cy7 anti-mouse CD45 clone 30-F11 (103115), APC anti-mouse/human CD11b clone M1/70 (101212), PE/Cyanine7 anti-mouse/human CD11b clone M1/70 (101216), PE/Dazzle 594 anti-mouse Ly6G clone 1A8 (127647), PerCP anti-mouse Ly-6G clone 1A8 (127653), PE anti-mouse F4/80 clone BM8 (123110), Brilliant Violet 605 anti-mouse Ly-6C clone HK1.4 (128035), APC anti-mouse I-A/I-E clone M5/114.15.2 (107614), Pacific Blue anti-mouse CD86 clone GL-1 (105022), APC/Cy7 anti-mouse CD86 clone GL-1 (105029), APC anti-mouse CD206 (MMR) clone C068C2 (141707), Brilliant Violet 421 anti-mouse CD206 clone C068C2 (141717), PE/Cy7 anti-mouse CD163 clone S15049F (156707), APC/Cy7 anti-mouse CD3e clone 1452C11 (100329), and Alexa Fluor 647 anti-mouse CD8a clone 53-6.7 (100727). TruStain FcX PLUS (anti-mouse CD16/32) clone S17011E (156603) was used in all immune infiltrate experiments to block Fc receptors. For immunogenicity post-MPS1i treatment, APC anti-mouse H-2Kb/H-2Db clone 28-8-6 (114613) was used. For IgG titer in tumor challenge surviving mice, the following BioLegend antibodies were used: PE anti-mouse IgG2a clone RMG2a-62 (407108, known to bind IgG2c as well) and APC anti-mouse IgG2b clone RMG2b-1 (406711). Primary antibody used in western blotting was anti-β-actin clone C4 (Santa Cruz sc 47778). Secondary antibody used in western blotting was HRP sheep anti-mouse IgG (GE Life Sciences NA931V).

## Mice

C57BL/6 mice (Jackson Laboratory 000664) were 6–12 weeks of age at the time of tumor challenges and for bone marrow harvesting, with the exception of second and third challenge experiments. Additionally, for re-challenge experiments, age-matched naïve mice were used. NSG mice (NOD. Cg-*Prkdc^{scid} Il2rg^{tm1Wjl}*/SzJ) aged 6 weeks were procured from and housed in the Perelman School of Medicine Stem Cell Xenograft Core. All experiments were performed in accordance with the protocols (#803177) approved by the Institutional Animal Care and Use Committee (IACUC) of the University of Pennsylvania.

## Drug treatments and micronuclei quantification

For MPS1i studies, the following chemical drugs were used: reversine (Cayman Chemical 10004412), AZ3146 (Cayman Chemical 19991), and BAY 12-17389 (Selleck Chemicals S8215). 24 hr prior to treatment, B16F10 cells were seeded in either 6-well or 12-well plates. For 6 wells, 20,000 cells were plated per well. For 12 wells, 2000 cells were plated per well. On the day of treatment, spent media was aspirated, and fresh media supplemented with 10% FBS, 1% P/S, and either MPS1i or DMSO vehicle control was added to each well. The concentration used for each treatment is listed in the figure legend. The volume of DMSO added was equal to the volume required for the highest MPS1i concentration for each experiment. All cells were treated for 24 hr, after which drug-containing spent media was aspirated. Cells were then washed with a full volume of PBS for 5 min. PBS was aspirated, and two repeat washes were performed. Cells were then allowed to recover from MPS1i treatment for an additional 48 hr, with a fresh media replacement 24 hr after the initial wash.

For imaging and micronuclei quantification, cells were fixed with 4% formaldehyde for 20 min after the 48 hr recovery period, stained with Hoechst for DNA, and later imaged on an Olympus IX inverted microscope with a 40×/0.6 NA objective. The Olympus IX microscope was equipped with a Prime sCMOS camera (Photometrics) and a pE-300 LED illuminator (CoolLED) and was controlled with MicroManager software v2. At least 200 B16F10 were imaged per individual well for micronuclei quantification.

## Bone marrow-derived macrophages

Bone marrow was harvested from the femurs and tibia of donor mice, lysed with ACK buffer (Gibco A1049201) to deplete red blood cells, and then cultured on Petri culture dishes for 7 days in Iscove's Modified Dulbecco's Medium (IMDM, Gibco 12440-053) supplemented with 10% FBS, 1% P/S, and 20 ng/mL recombinant mouse macrophage colony-stimulating factor (M-CSF, BioLegend 576406). 72 hr after initial plating, one whole volume of fresh IMDM supplemented 10% FBS, 1% P/S, and 20 ng/mL M-CSF was added. After 7 days of differentiation, spent media was removed, BMDMs were gently washed once with PBS, and fresh IMDM supplemented with 10% FBS, 1% P/S, and 20 ng/mL M-CSF was added.

## Conditioned media treatment of BMDMs

BMDMs that had successfully undergone 7 days of differentiation in 20 ng/mL M-CSF were used. Spent media was removed, BMDMs were gently washed once with PBS, and fresh IMDM supplemented with 10% FBS, 1% P/S, and 20 ng/mL M-CSF was added. Then, conditioned media from B16F10 cells treated with either MPS1i or DMSO was collected. Conditioned media was centrifuged for 5 min at 300 × $g$ to remove any cellular debris. The supernatant was collected and then supplemented with 5% FBS, 1% P/S, and 20 ng/mL M-CSF. One whole volume of conditioned media was added to BMDMs with fresh media. 24 hr after treatment, BMDMs were detached using 0.05% Trypsin and processed for single-cell RNA-sequencing.

## In vitro phagocytosis

For 2D phagocytosis assays, BMDMs were detached using 0.05% Trypsin and re-plated in either 6-well or 12-well plates, at a density of $1.8 \times 10^4$ cells per $cm^2$ in IMDM supplemented with 10% FBS, 1% P/S, and 20 ng/mL M-CSF. After 24 hr elapsed, BMDMs were labeled with 0.5 µM CellTracker DeepRed dye (Invitrogen C34565), according to the manufacturer's protocol. Following staining, BMDMs were washed and incubated in serum-free IMDM supplemented with 0.1% (wt/vol) BSA and 1% P/S. B16F10 cells were labeled with carboxyfluorescein diacetate succinimidyl ester (Vybrant CFDA-SE Cell Tracer, Invitrogen V12883), also according to the manufacturer's protocol. B16F10 cells were detached and opsonized with 10 µg/mL anti-Tyrp1, with 10 µg/mL mouse IgG2a isotype control antibody, or 5% (vol/vol) mouse serum in 1% BSA. Opsonization was allowed for 30–45 min on ice. Opsonized B16F10 suspensions were then added to BMDMs at an ~2:1 ratio and incubated at 37°C and 5% $CO_2$ for 2 hr. Non-adherent cells were removed by gently washing with PBS. For imaging, cells were fixed with 4% formaldehyde for 20 min and later imaged on an Olympus IX inverted microscope with a 40×/0.6 NA objective. The Olympus IX microscope was equipped with a Prime sCMOS camera (Photometrics) and a pE-300 LED illuminator (CoolLED) and was controlled with MicroManager software v2. At least 300 macrophages were imaged per individual well for calculation of phagocytosis.

## 3D tumoroid formation and phagocytosis

Briefly, non-TC-treated 96-well U-bottom plates were treated with 100 µL of anti-adherence rinsing solution (StemCell Technologies 07010) for 1 hr. The cells were then washed with 100 µL of complete RPMI 1640 cell culture media. This generated surfaces conducive to generating tumoroids and preventing cells from adhering to the well bottom during experiments. B16F10 were detached by brief trypsinization, resuspended at a concentration of $1\times10^4$ cells per mL in complete RPMI 1640 cell culture media (10% FBS, 1% P/S) with 50 µM β-mercaptoethanol (Gibco 21985023). 100 µL of this cell suspension was added to each well such that each tumoroid initially started with approximately $1\times10^3$ cells. Aggregation of B16F10 cells was confirmed 24 hr later by inspection under microscopy. Upon confirmation of tumoroid formation, BMDMs were labeled with 0.5 µM CellTracker DeepRed dye (Invitrogen C34565), according to the manufacturer's protocol. BMDMs were then detached by brief trypsinization and gentle scraping and resuspended in complete RPMI 1640 cell culture media at a concentration that would allow for delivery of $3\times10^3$ BMDMs to each individual tumoroid culture. The BMDM cell suspension was also supplemented with 120 ng/mL M-CSF and antibodies (either anti-Tyrp1 or mouse IgG2a isotype control) such that delivery of 20 µL of this suspension to each individual tumoroid culture result in final concentrations of 20 ng/mL M-CSF and 20 µg/mL of antibody. For tumoroid studies in which mouse convalescent serum was used, the BMDM cell suspension was supplemented with 120 ng/mL M-CSF and serum such that delivery of 20 µL of this suspension to each individual tumoroid culture resulted in final concentrations of 20 ng/mL M-CSF and a final mouse serum concentration of 1:200. Tumoroids were imaged on an Olympus IX inverted microscope with a 4×/0.13 NA objective.

For macrophage polarization tumoroid experiments (*Figure 2B–C*), B16F10 tumoroids were prepared in the same manner as described above, with the exception that no anti-Tyrp1 or mIgG2a was added. After 5 days had elapsed for the experiment, tumoroids were dissociated by brief trypsinization and stained with conjugated antibodies targeting MHCII, CD86, CD163, and CD206, following the manufacturer's protocol. Cells were then run on a BD LSRII (Benton Dickinson) flow cytometer. Data were analyzed with FCS Express 7 software (De Novo Software).

## In vivo tumor growth

B16F10 cells cultured in DMEM growth media were detached by brief trypsinization, washed twice with PBS, and resuspended at $2\times10^6$ cells per mL. Cell suspensions were kept on ice until injection. All subcutaneous injections were performed on the right flank while mice were anesthetized under isoflurane. Fur on the injection site was wet slightly with a drop of 70% ethanol and brushed aside to better visualize the skin for injection beneath the skin of a 100 µL suspension of cancer cells. For assessing immune infiltrates in early stages of tumor engraftment, when tumors are still small, we used a relatively high number of tumor cells (500,000 cells in *Figure 1D* and *Figure 2F and G*) to achieve sufficient cell numbers after dissociating the tumors, particularly for the slow-growing MPS1i-treated tumors. More specifically, with dissection, collagenase treatment, passage through a filter to remove clumps, we would lose many cells, and yet needed 100,000 viable cells or more for bulk RNA-sequencing suspensions and for flow cytometry measurements. For all other studies, 200,000 cancer cells were injected, and for subsequent treatments, mice received either intravenous or intraperitoneal injections of anti-Tyrp1 clone TA99 or mouse IgG2a isotype control clone C1.18.4 (250 µg antibody in 100 µL PBS) on days 4, 5, 7, 9, 11, 13, and 15 post-tumor challenge. Intravenous injections were done via the lateral tail vein. Tumors were monitored by palpation and measured with digital calipers. The projected area was roughly elliptical and was calculated as $A = \pi/4 \times L \times W$, where L is the length along the longest axis and W is the width measured along the perpendicular axis. For our studies, a projected area of 125 mm$^2$ was considered terminal burden for survival analyses. Mice were humanely euthanized following IACUC protocols if tumor size reached 2.0 cm on either axis, if tumor reached a projected area greater than 200 mm$^2$ or if a tumor was ulcerated.

## Adoptive cell transfers

Fresh bone marrow was harvested as described in the Bone marrow-derived macrophages section of Materials and methods. Marrow cells were counted on a hemocytometer and resuspended to a concentration of $8\times10^7$ cells per mL in 5% (vol/vol) FBS/PBS. To block SIRPα, cells were then incubated with anti-SIRPα clone P84 (18 µg/mL) for 1 hr at room temperature on a rotator. After the incubation

period elapsed, cells were centrifuged at 300 × g for 5 min, washed with PBS, and then centrifuged once more at 300 × g for 5 min to remove any unbound anti-SIRPα. Cells were again resuspended to a concentration of 8×10⁷ cells per mL in 2% (vol/vol) FBS/PBS, with or without anti-Tyrp1 clone TA99 (1 mg/mL). Marrow cells were then injected intravenously (2×10⁷ cells in 250 µL per mouse) into tumor-bearing mice. All adoptive transfers were done 4 days post-challenge. Control data were adapted from *Dooling et al., 2023*, to minimize mice for experiments, since the cited study establishes proper benchmarks for comparison and also finds that these control conditions minimally (if at all) improve survival.

## Serum collection and IgG titer quantification

Blood was drawn retro-orbitally from mice anesthetized under isoflurane, using heparin- or EDTA-coated microcapillary tubes. Collected blood was allowed to clot for 1 hr at room temperature in a microcentrifuge tube. The serum was separated from the clot by centrifugation at 1500 × g and stored at –20°C for use in flow cytometry and phagocytosis assays.

For IgG titer quantification, B16F10 cells were detached by trypsinization and incubated with 5% (vol/vol) mouse serum in 1% BSA. Opsonization was allowed for 30–45 min on ice. After the incubation period elapsed, cells were centrifuged at 300 × g for 5 min, washed once with PBS, centrifuged again at 300 × g for 5 min, and then resuspended in 0.1% (wt/vol) BSA with both PE anti-mouse IgG2a/c clone RMG2a-62 and APC anti-mouse IgG2b clone RMG2b-1 (see Antibodies section for more information). Anti-IgG-conjugated antibody incubation occurred for 30–45 min, after which cells were centrifuged at 300 × g for 5 min, washed once with PBS, centrifuged again at 300 × g for 5 min, and then resuspended in 5% (vol/vol) FBS/PBS. Cells were run on a BD LSRII (Benton Dickinson) flow cytometer.

## Western blotting

Lysate was prepared from B16F10 cells using RIPA buffer containing 1× protease inhibitor cocktail (Millipore Sigma P8340) and boiled in 1× NuPage LDS sample buffer (Invitrogen NP0007) with 2.5% β-mercaptoethanol. Proteins were separated by electrophoresis in NuPage 4–12% Bis-Tris gels run with 1× MOPS buffer (Invitrogen NP0323) and transferred to an iBlot nitrocellulose membrane (Invitrogen IB301002). The membranes were blocked with 5% (wt/vol) non-fat milk in Tris-buffered saline (TBS) plus Tween-20 (TBST) for 1 hr and stained with 5% (vol/vol) mouse serum overnight at 4°C with agitation. The membranes were washed with TBST and incubated with 1:500 secondary antibody conjugated with horseradish peroxidase (HRP) in 5% (wt/vol) milk in TBST for 1 hr at room temperature with agitation. The membranes were then washed again three times with TBST. Membranes probed with HRP-conjugated secondary antibody were developed a 3,3',5,5'-teramethylbenzidine substrate (Millipore Sigma T0565). Developed membranes were scanned and then processed with ImageJ.

## Immune infiltrate analysis of tumors

For day 5 post-challenge measurements: If mice required anti-Tyrp1 treatment, mice received a single dose of intravenously delivered anti-Tyrp1 or mouse IgG2a isotype control 4 days (96 hr) post-tumor challenge. 24 hr later, mice were humanely sacrificed. Otherwise, mice were sacrificed 5 days post-tumor challenge. For immune analysis of second challenge non-survivors: Mice were humanely euthanized when tumor burden reached >150 mm². 

Tumors from euthanized mice were excised and placed into 5% (vol/vol) FBS/PBS. Tumors were then disaggregated using Dispase (Corning 354235) supplemented with 4 mg/mL of collagenase type IV (Gibco 17104-019) and DNAse I (Millipore Sigma, 101041159001) for 30–45 min (until noticeable disaggregation) at 37°C, centrifuged for 5 min at 300 × g, and resuspended in 1 mL of ACK lysis buffer for 12 min at room temperature. Samples were centrifuged for 5 min at 300 × g, washed once with PBS, and then resuspended in 5% (wt/vol) BSA/PBS for 20 min. After 20 min elapsed, fluorophore-conjugated antibodies to immune markers were added to each cell suspension. The following markers were used for analysis: for macrophages, CD45, CD11b, F4/80, Ly-6C, Ly-6G, CD86, CD206, MHCII; for T cells, CD45, CD3e, CD8a. Antibody binding occurred for 30 min while samples were kept on ice and covered from light. Samples were then centrifuged for 5 min at 300 × g, washed once with PBS, and resuspended in FluoroFix Buffer (BioLegend, 422101) for 1 hr at room temperature prior to analysis on a BD LSRII (Benton Dickinson) flow cytometer. For day 5 post-challenge measurements,

100,000–200,000 live cells were collected. For in vivo tumor infiltrate studies in re-challenged mice, 10 million live cells were collected. Data were analyzed with FCS Express 7 software (De Novo Software).

## Bulk RNA-sequencing

RNA library was constructed using NEBNext Ultra II RNA Library Prep Kit for Illumina and NEBNext Multiplex Oligos for Illumina (E7770S, E7335) per the manufacturer's instructions. The library prepared was processed at the Next Generation Sequencing Core at the University of Pennsylvania (12-160, Translational Research Center) using NovaSeq 6000, 100 cycles (Illumina). The reads were aligned to mouse reference, mm10 (GENCODE vM23/Ensembl 98) using star alignment. Cell count matrix was generated and imported to RStudio for downstream analysis. Package 'DESeq2' (v1.32.0) was used for normalization and differential expression analysis. Package 'fgsea' (v1.18.0) was used for GSEA. Additional exploratory data analysis was then done using either RStudio or Python 3.8.

## Single-cell RNA-sequencing

RNA libraries were prepared using the 10x Genomics Chromium Single Cell Gene Expression kit (v3.1, single index, PN-1000128; PN-1000127; PN-1000213) per the manufacturer's instructions. The libraries were sequenced at the Next Generation Sequencing Core using NovaSeq 6000, 100 cycles (Illumina). Raw base call (BCL) files were analyzed using CellRanger (version 5.0.1) to generate FASTQ files, and the 'count' command was used to generate raw count matrices aligned to mm10 (GENCODE vM23/Ensembl 98). Cells were filtered to make sure that they expressed a minimum of 1400 genes with less than 15% mitochondrial content. Data was normalized using the 'LogNormalize'' method from the Seurat package. Differential expression analysis was performed using the 'FindAllMarkers'' command, and the output was used for GSEA.

## InferCNV

Count matrix of single-cell RNA-sequencing results was used as input for InferCNV object construction (1.7.1) (InferCNV of the Trinity CTAT Project, see the following for more information: https://github.com/broadinstitute/inferCNV, Copy archived by *infercnv, 2024*). Gene position files were created for GRCm38. Single-cell RNA-sequencing data of DMSO-treated B16F10 were used as reference for copy number profile construction. Cell types were annotated either manually or using the package 'SingleR' (v1.6.1). For manual annotation, cells were clustered and assigned cell types based on the expression of cell type-specific signature genes (SI.5a). Denoised results from InferCNV were used as the input (' infercnv.observations.txt'). The averaged copy number of each chromosome segment was calculated, and the difference between each cell's copy number and the overall mean at each segment was calculated. The deviation was summed across the entire chromosome to obtain the distribution of the deviation. Cells sharing an absolute deviation that is more than 2.5 times standard deviation away from the distribution peak were marked as outliers for a certain chromosome in question.

## Transcriptomic gene sets

For analyzing both bulk and single-cell RNA-sequencing datasets, either hallmark gene sets (from the Human Molecular Signatures Database) or customized gene sets were used. Customized gene sets were used exclusively for macrophage-associated analyses and were made by combining gene sets from the following: *Ahn et al., 2018*; *Cerezo-Wallis et al., 2020*; *Cunha et al., 2018*; *Mujal et al., 2022*; *Perry et al., 2019*; *Zhou et al., 2020*.

## Statistical analysis and curve fitting

Statistical analyses and curve fitting were performed in GraphPad Prism 9.4. Details for each analysis are provided in the figure legends. For differential gene expression analysis for both bulk RNA-sequencing and single-cell RNA-sequencing datasets, statistical analyses were done using either RStudio 2022.02.3+492 or Python 3.8.

## Code availability

Sequencing data were analyzed using RStudio 2022.02.3+492 and the Seurat package. Additional analyses after normalization were done on either RStudio or Python 3.8. No new code central to the conclusions of this study was developed.

## Acknowledgements

This work was supported by funding from the following sources: NIH U01 CA254886 (DED), P01 CA265794 (DED), NSF GRFP DGE-1845298 (BHH, JCA, MPT), and NIH F32 CA228285 (LJD). The authors acknowledge the following University of Pennsylvania core facilities: Cell Center Stockroom, the Penn Cytomics and Cell Sorting Resource Laboratory, the Penn Genomic and Sequencing Core, and the Cell and Development Biology Microscopy Core.

## Additional information

### Funding

| Funder | Grant reference number | Author |
| --- | --- | --- |
| National Institutes of Health | U01 CA254886 | Dennis E Discher |
| National Institutes of Health | P01 CA265794 | Dennis E Discher |
| National Science Foundation | GRFP DGE-1845298 | Brandon H Hayes<br>Jason C Andrechak<br>Michael P Tobin |
| National Institutes of Health | F32 CA228285 | Lawrence J Dooling |

The funders had no role in study design, data collection and interpretation, or the decision to submit the work for publication.

### Author contributions

Brandon H Hayes, Conceptualization, Formal analysis, Funding acquisition, Investigation, Visualization, Methodology, Writing - original draft, Writing - review and editing; Mai Wang, Data curation, Formal analysis, Investigation, Visualization, Methodology; Hui Zhu, Steven H Phan, Alexander H Chang, Formal analysis, Investigation; Lawrence J Dooling, Conceptualization, Formal analysis, Funding acquisition, Investigation, Methodology; Jason C Andrechak, Funding acquisition, Investigation; Michael P Tobin, Nicholas M Ontko, Tristan Marchena, Investigation; Dennis E Discher, Conceptualization, Resources, Formal analysis, Funding acquisition, Methodology, Writing - review and editing

### Author ORCIDs

Brandon H Hayes ● http://orcid.org/0000-0002-9099-201X
Lawrence J Dooling ● http://orcid.org/0000-0002-1688-2066
Michael P Tobin ● http://orcid.org/0000-0002-9607-2062
Dennis E Discher ● http://orcid.org/0000-0001-6163-2229

### Ethics

All experiments were performed in accordance with protocols (#803177) approved by the Institutional Animal Care and Use Committee (IACUC) of the University of Pennsylvania.

Reviewer #2 (Public review): https://doi.org/10.7554/eLife.88054.3.sa1
Author response https://doi.org/10.7554/eLife.88054.3.sa2

## Additional files

### Supplementary files

• MDAR checklist

### Data availability

Sequencing data have been deposited in GEO Gene Expression Omnibus under accession codes GSE261556 and GSE261555. All other data are available within the article and its supplementary information.

The following datasets were generated:

| Author(s) | Year | Dataset title | Dataset URL | Database and Identifier |
|---|---|---|---|---|
| Phan SH, Hayes BH, Wang M, Discher DE | 2024 | Chromosomal instability can favor macrophage-mediated immune response and induce a broad, vaccination-like anti-tumor IgG | https://www.ncbi.nlm.nih.gov/geo/query/acc.cgi?acc=GSE261556 | NCBI Gene Expression Omnibus, GSE261556 |
| Phan SH, Hayes BH, Wang M, Discher DE | 2024 | Chromosomal instability can favor macrophage-mediated immune response and induce a broad, vaccination-like anti-tumor IgG | https://www.ncbi.nlm.nih.gov/geo/query/acc.cgi?acc=GSE261555 | NCBI Gene Expression Omnibus, GSE261555 |

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

# Appendix 1

## Appendix 1—key resources table

| Reagent type (species) or resource | Designation | Source or reference | Identifiers | Additional information |
|---|---|---|---|---|
| Strain, strain background *Mus musculus* | C57BL/6J | Jackson Laboratory | Stock# 000664; RRID:IMSR_JAX:000664 | Sex: Male (because B16 cell line is male derived) |
| Strain, strain background *Mus musculus* (male) | NOD.Cg-Prkdc$^{scid}$ Il2rg$^{tm1Wjl}$/SzJ | Perelman School of Medicine Stem Cell Xenograft Core | RRID:SCR_010035 | Sex: Male |
| Cell line (*Mus musculus*) | B16F10 | ATCC | Cat# CRL-6475; RRID:CVCL_0159 | Mouse melanoma |
| Cell line (*Mus musculus*) | B16F10 CD47 KO | PMID: 31964705 | CD47 KO | *Hayes et al., 2020* |
| Cell line (*Mus musculus*) | B16F10 sgCtrl | PMID: 31964705 | sgCtrl | *Hayes et al., 2020* |
| Biological sample (*Mus musculus*) | C57BL/6J bone marrow cells | This paper | | Flushed from femurs, tibias; differentiation into bone marrow-derived macrophages (BMDMs) or adoptive transfer |
| Biological sample (*Mus musculus*) | C57BL/6J serum | This paper | | Retro-orbital blood collected from convalescent or naïve mice, clotted for 1 hr, separated by centrifugation |
| Antibody | InVivoMab mouse anti-human/mouse Tryp1 (clone TA99) | BioXCell | Cat# BE0151; RRID:AB_10949462 | (1:1000) Opsonization (10–20 µg/mL) In vivo (250 µg/100 µL per dose) |
| Antibody | InVivoMab mouse IgG2a isotype control (clone C1.18.4) | BioXCell | Cat# BE0085; RRID:AB_1107771 | (1:1000) (10–20 µg/mL) In vivo (250 µg/100 µL per dose) |
| Antibody | Ultra-LEAF rat anti-mouse SIRPα (clone P84) | BioLegend | Cat# 144036; RRID:AB_2832517 | Blockade (18 µg/mL=1:50) |
| Antibody | InVivoMab rat anti-mouse CD47 (clone MIAP301) | BioXCell | Cat# BE0270; RRID:AB_2687793 | (1:500) Flow cytom. (20 µg/mL) |
| Antibody | Alexa Fluor 647 anti-mouse IgG [H+L] (donkey polyclonal) | Thermo Fisher Invitrogen | Cat# A-31571; RRID:AB_162542 | Flow cytom. (10 µg/mL=1:200) |
| Antibody | Alexa Fluor 647 anti-rat IgG [H+L] (goat polyclonal) | Thermo Fisher Invitrogen | Cat# A-21247; RRID:AB_141778 | Flow cytom. (10 µg/mL=1:200) |
| Antibody | Horse radish peroxidase sheep anti-mouse [H+L], polyclonal | GE Life Sciences | Cat# NA931V | WB (1:500) |
| Antibody | Mouse anti-β-actin (clone C4) | Santa Cruz | Cat# sc47778; RRID:AB_626632 | WB (1:1000) |

*Appendix 1 Continued on next page*

*Appendix 1 Continued*

| Reagent type (species) or resource | Designation | Source or reference | Identifiers | Additional information |
|---|---|---|---|---|
| Antibody | Brilliant Violet 650 rat anti-mouse CD45 (clone 30-F11) | BioLegend | Cat# 103151; RRID:AB_2565884 | Flow cytom. (2.5 µg/mL=1:80) |
| Antibody | Brilliant Violet 785 rat anti-mouse CD45 (clone 30-F11) | BioLegend | Cat# 103149; RRID:AB_2564590 | Flow cytom. (5 µg/mL=1:40) |
| Antibody | APC/Cyanine7 rat anti-mouse CD45 (clone 30-F11) | BioLegend | Cat# 103115; RRID:AB_312980 | Flow cytom. (2.5 µg/mL=1:80) |
| Antibody | APC rat anti-mouse/human CD11b (clone M1/70) | BioLegend | Cat# 101212; RRID:AB_312795 | Flow cytom. (2.5 µg/mL=1:80) |
| Antibody | PE/Cyanine7 rat anti-mouse/human CD11b (clone M1/70) | BioLegend | Cat# 101216; RRID:AB_312799 | Flow cytom. (2.5 µg/mL=1:80) |
| Antibody | PE/Dazzle 594 rat anti-mouse Ly6G (clone 1A8) | BioLegend | Cat# 127647; RRID:AB_2566318 | Flow cytom. (5 µg/mL=1:40) |
| Antibody | PerCP rat anti-mouse Ly6G (clone 1A8) | BioLegend | Cat# 127653; RRID:AB_2616998 | Flow cytom. (2.5 µg/mL=1:80) |
| Antibody | PE rat anti-mouse F4/80 (clone BM8) | BioLegend | Cat# 123110; RRID:AB_893486 | Flow cytom. (10 µg/mL=1:20) |
| Antibody | Brilliant Violet 605 rat anti-mouse Ly-6C (clone HK1.4) | BioLegend | Cat# 128035 RRID:AB_2562352 | Flow cytom (5 µL/100 µL=1:20) |
| Antibody | APC rat anti-mouse I-A/I-E (clone M5/114.15.2) | BioLegend | Cat# 107614; RRID:AB_313329 | Flow cytom. (2.5 µg/mL=1:80) |
| Antibody | Pacific Blue rat anti-mouse CD86 (clone GL-1) | BioLegend | Cat# 105022; RRID:AB_493466 | Flow cytom. (10 µg/mL=1:50) |
| Antibody | APC/Cyanine7 rat anti-mouse CD86 (clone GL-1) | BioLegend | Cat# 105029; RRID:AB_2074993 | Flow cytom. (2.5 µg/mL=1:80) |
| Antibody | APC rat anti-mouse CD206 (clone C068C2) | BioLegend | Cat# 141707; RRID:AB_10896057 | Flow cytom. (5 µg/mL=1:40) |
| Antibody | Brilliant Violet 421 rat anti-mouse CD206 (clone C068C2) | BioLegend | Cat# 141717; RRID:AB_2562232 | Flow cytom. (5 µL/100 µL=1:20) |
| Antibody | PE/Cyanine7 rat anti-mouse CD163 (clone S15049F) | BioLegend | Cat# 156707; RRID:AB_2910324 | Flow cytom. (2.5 µg/mL=1:80) |
| Antibody | APC/Cyanine7 Armenian hamster anti-mouse CD3e (clone 145-2C11) | BioLegend | Cat# 100329; RRID:AB_1877171 | Flow cytom. (5 µg/mL=1:40) |
| Antibody | Alexa Fluor 647 rat anti-mouse CD8a (clone 53-6.7) | BioLegend | Cat# 100727; RRID:AB_493424 | Flow cytom. (2.5 µg/mL=1:200) |

*Appendix 1 Continued on next page*

*Appendix 1 Continued*

| Reagent type (species) or resource | Designation | Source or reference | Identifiers | Additional information |
|---|---|---|---|---|
| Antibody | APC mouse anti-mouse H-2kb/H-2Db (clone 28-8-6) | BioLegend | Cat# 114613; RRID:AB_2750193 | Flow cytom. (5 µg/mL=1:40) |
| Antibody | PE rat anti-mouse IgG2a (clone RMG2a-62) | BioLegend | Cat# 407108; RRID:AB_10549974 | Flow cytom. (5 µg/mL=1:40) |
| Antibody | APC rat anti-mouse IgG2b (clone RMG2b-1) | BioLegend | Cat# 406711; RRID:AB_2750277 | Flow cytom. (2.5 µg/mL=1:80) |
| Antibody | TruStain FcX PLUS rat anti-mouse CD16/32 (clone S17011E) | BioLegend | Cat# 156603; RRID:AB_2783137 | Flow cytom. Fc block (2.5 µg/mL=1:200) |
| Peptide, recombinant protein | Macrophage colony-stimulating factor (M-CSF) | BioLegend | Cat# 576406 | BMDM differentiation (20 ng/mL) |
| Commercial assay or kit | Vybrant CFDA-SE Cell Trace | Thermo Fisher Invitrogen | Cat# V12883 | |
| Commercial assay or kit | CellTracker DeepRed | Thermo Fisher Invitrogen | Cat# C34565 | |
| Commercial assay or kit | NEBNext Ultra II RNA Library Prep Kit | NEB | Cat# E7770S | |
| Sequence-based reagent | NEBNext Multiplex Oligos for Illumina | NEB | Cat# E7335 | |
| Commercial assay or kit | Chromium Single Cell Gene Expression kit | 10x Genomics | Cat# PN-1000128; Cat# PN-1000127; Cat# PN-1000213 | |
| Chemical compound, drug | Reversine | Cayman Chemical | Cat# 10004412 | MPS1 inhibitor |
| Chemical compound, drug | AZ3146 | Cayman Chemical | Cat# 19991 | MPS1 inhibitor |
| Chemical compound, drug | BAY 12-17389 | Selleck Chemicals | Cat# S8215 | MPS1 inhibitor |
| Software, algorithm | Prism v9.4 | GraphPad | RRID:SCR_002798 | |
| Software, algorithm | FCS Express 7 | De Novo Software | | |
| Other | Anti-adherence rinsing solution | StemCell Technologies | Cat# 07010 | Surface treatment for tumoroid studies |
| Other | ACK lysing buffer | Thermo Fisher Gibco | Cat# A1049201 | Bone marrow and tumor red blood cell lysis |
| Other | Dispase | Corning | Cat# 354235 | Tumor disaggregation |
| Other | DNAse I | Millipore Sigma | Cat# 101041159001 | Tumor disaggregation |
| Other | Collagenase type IV | Thermo Fisher Gibco | Cat# 17104-019 | Tumor disaggregation (4 mg/mL) |

