## [Editor Report · eLife assessment]

The authors provide **compelling** evidence that MSP1 inhibition (leading to chromosomal instability or CIN in the cancer cells) increases phagocytosis and that tumors with CIN respond better to macrophage therapeutics. In this **important** study, they demonstrate particularly impressive survival rates for mouse models of CIN B16 tumors treated with adoptively transferred macrophages, CD47-SIRPα blockade, and anti-Tyrp1 IgG.

---

## [Referee Report · Reviewer #2 (Public review)]

Harnessing macrophages to attack cancer is an immunotherapy strategy that has been steadily gaining interest. Whether macrophages alone can be powerful enough to permanently eliminate a tumor is a high-priority question. In addition, the factors making different tumors more vulnerable to macrophage attack have not been completely defined. In this paper, the authors find that MSP1 inhibition, most notable for causing chromosomal instability (CIN), in cancer cells improves the effect of macrophage targeted immunotherapies. They demonstrate that MSP1 inhibited tumors secrete factors that polarize macrophages to a more tumoricidal fate through several methods. The most compelling experiment is transferring conditioned media from MSP1 inhibited and control cancer cells, then using RNAseq to demonstrate that the MSP1-inhibited conditioned media causes a shift towards a more tumoricidal macrophage phenotype. In mice with MSP1 inhibited (CIN) B16 melanoma tumors, a combination of CD47 knockdown and anti-Tyrp1 IgG is sufficient for long term survival in nearly all mice. This combination is a striking improvement from conditions without CIN.

Like any interesting paper, this study leaves several unanswered questions. First, how do CIN tumors repolarize macrophages? The authors demonstrate that conditioned media is sufficient for this repolarization, implicating secreted factors, but the specific mechanism is unclear. The main caveat of the study is that chromosomal instability is driven by MSP1 inhibition in all the experiments, leaving open the possibility that some effects are due to MSP1 inhibition specifically rather than CIN more generally. To specifically connect CIN and macrophage repolarization, future studies will need to examine tumors with CIN unrelated to MSP1 inhibition to determine if these are also able to repolarize macrophages.

Overall, this is a thought-provoking study that will be of broad interest to many different fields including cancer biology, immunology and cell biology.

---

## [Author Response]

[The following is the authors’ response to the current reviews.]

**Reviewer #2 (Recommendations For The Authors):**

We sincerely appreciate the time and efforts of the Reviewer.

In light of your data showing that the IgG response is similar with and without CIN, it would be good to drop "and induce abroad, vaccination-like anti-tumor IgG response". This suggests a direct connection between CIN and the IgG response.In my opinion, the shorter title is equally strong and more correct.

We edited this phrase in the originally submitted title for accuracy:

“Chromosomal instability induced in cancer can enhance macrophage-initiated immune responses that include anti-tumor IgG”

I agree that inducing CIN through other means can be left for a different study but in that case the abstract should moredirectly mention MSP1 inhibition since that is how CIN is always induced. Perhaps line 18: CIN is induced by MSP-1inhibition in poorly immunogenic....

Done as requested:

“…Here, CIN is induced in poorly immunogenic B16F10 mouse melanoma cells using spindle assembly checkpoint MPS1 inhibitors…”

[The following is the authors’ response to the original reviews.]

**eLife assessment**
This study highlights a valuable finding that chromosomal instability can change immunes responses, in particular macrophages behaviours. The convincing results showing that the use of CD47 targeting and anti-Tyrp1 IgG can overcome changes in immune landscape in tumors and prolong survival of tumor-bearing mice. These findings reveal a new exciting dimension on how chromosomal instability can influence immune responses against tumor.

We thank the Editors for their enthusiasm and appreciation for this work. We also want to highlight our thanks for their careful reading, support, and patience while handling this manuscript. While this work provides useful insight into potential therapeutic implications of chromosomal instability in the macrophage immunotherapy field, we also hope it elucidates some novel basic science to further explore how chromosomal instability has such interesting effects on the immune system.

**Public Reviews:**

**Reviewer #1 (Public Review):**
The manuscript by Hayes et al. explored the potential of combining chromosomal instability with macrophage phagocytosis to enhance tumor clearance of B16-F10 melanoma. However, the manuscript suffers from substandard experimental design, some contradictory conclusions, and a lack of viable therapeutic effects.The authors suggest that early-stage chromosomal instability (CIN) is a vulnerability for tumorigenesis, CD47-SIRPa interactions prevent effective phagocytosis, and opsonization combined with inhibition of the CD47-SIRPa axis can amplify tumor clearance. While these interactions are important, the experimental methodology used to address them is lacking.
**Reviewer #1 (Recommendations For The Authors):**
First, early stages of the tumor are essentially being defined as before implantation. In all cases, the tumor cells were pre-treated with MPS1i or had a genetic knockout of CD47. This makes it difficult to see how this would translate clinically.

We greatly appreciate the Reviewer’s interest in the topic and its potential, but our manuscript makes no claims of immediate clinical translation. Chromosomal instability (CIN) studies have to date not yet discovered or described whether and how CIN can affect macrophage function. To our knowledge, this is the first study to begin such characterizations with various MPS1i drugs to induce CIN. Many variations of the approach can be envisioned for future studies.

Our Results include some key studies of cancer cells with wildtype levels of CD47- including in vivo tumor elimination (Fig.3E). Nonetheless, we do conduct some of our studies in a CD47 knockout context to remove this “brake” that generally impedes phagocytosis, with our goal being to better understand how CIN affects phagocytosis. As cited to some extent in our Introduction, there are many efforts in clinical trials to disrupt this macrophage checkpoint and others focused on macrophage immunotherapy. Whether CIN can be induced by clinically translatable drugs and specifically in cancer cells is beyond the scope of our studies.

I would like to see the amount of CIN that occurs in WT B16F10 over the course of tumorigenesis (ie longer than 5 days). This is because I would assume that CIN would eventually occur in the WT B16F10 regardless of whether MPS1i is being given. And if that's the case, then the initiation of CIN at day 10 after implantation (for example) would still be considered "early stage" CIN. If the therapy is then initiated at this point, does the effect remain? Or put differently, how would the authors propose to induce the appropriate level of CIN in an established tumor? Why is pretreatment necessary?

Untreated B16F10 cells fail to produce micronuclei over 12 days compared to MPS1i treated cells – as shown in a newly added panel in Fig. S1:

**Author response image 1. sa2fig1:** 

This helps support our decision to pre-treat cells with MPS1i to stimulate genomic instability and is described in the first section of Results:

“…we saw >10-fold increases of micronuclei over the cell line’s low basal level (~1% of cells), and two other MPS1i inhibitors AZ3146 and BAY12-17389 confirm such effects (Fig. S1A). Micronuclei-positive cells can persist up to 12 days after treatment (Fig. S1B), while control cells maintain the low basal levels. The results suggest pre-treatment with MPS1i can simulate CIN in an experimental context even for 1-2 weeks, which may not typically occur at the same frequency during early tumor growth.

It is known that PD-1 expression inhibits tumor-associated macrophage phagocytosis (Nature, 2017). Does MSP1i (sic) treatment affect the population of PD-1+ tumor macrophages in vivo?

We thank the Reviewer for bringing up an interesting point.

Using the same tumor RNA-seq data that was used for Fig.1E, a heatmap of expression of PD-1 (gene Pdcd1) shows no consistent trend with MPS1i:

We also examined whether the secretome from CIN-afflicted cancer cells affect PD-1 expression in cultured macrophages, but we did not register any reads from our single-cell RNA-sequencing experiment for Pdcd1 in any of the macrophage clusters from Fig. 1H.

**Author response image 3. sa2fig3:** 

The Discussion section now includes a statement on this topic:

“…B16F10 tumors are poorly immunogenic, do not respond to either anti-CD47 or anti-PD-1/PDL1 monotherapies, and show modest and variable cure rates (~20-40%; Dooling et al., 2023; Hayes et al., 2023) even when macrophages have been made maximally phagocytic according to notions above. We should note here that our whole-tumor RNA-seq data (Fig.1E) shows expression of PD-1 (gene Pdcd1) follows no consistent trend upon MPS1i treatment, and that Pdcd1 was not detected in our scRNA-seq data for macrophage cultures (Fig.1G) – motivating further study.”

The authors must explain how the proposed therapy works since MPS1i increases tumor (cell) size, making it difficult for macrophages to phagocytose the tumor cells. It also reduces or suppresses Tyrp1 expression on the cancer cells, making it harder to opsonize. Since these were two main points for the rationale of this study, the authors need to reconcile them.

We appreciate this comment and have re-organized this Results section to try to minimize confusion:

CIN-afflicted, CD47-knockout tumoroids are eliminated by Macrophages

To assess functional effects of macrophage polarization, we focused on a 3D “immuno-tumoroid” model in which macrophage activity can work (or not) over many days against a solid proliferating mass of cancer cells in non-adherent roundbottom wells (Fig. 2A) (Dooling et al., 2023). We used CD47 knockout (KO) B16F10 cells, which removes the inhibitory effect of CD47 on phagocytosis, noting that KO does not perturb surface levels of Tyrp1, which is targetable for opsonization with anti-Tyrp1 (Fig. S2A). BMDMs were added to pre-assembled tumoroids at a 3:1 ratio, and we first assessed surface protein expression of macrophage polarization markers. Consistent with our whole-tumor bulk RNA-sequencing and also single-cell RNA-sequencing of BMDM monocultures (Fig. 1E, 1I-J), BMDMs from immunotumoroids of MPS1i-treated B16F10 showed increased surface expression of M1-like markers MHCII and CD86 while showing decreased expression of M2-like markers CD163 and CD206 (Fig. 2B-C). Although these macrophages seemed poised for anticancer activity, the cancer cells showed decreased binding of anti-Tyrp1 (Fig. S2B) and ~20% larger size in flow cytometry (Fig. S2C). The latter likely reflects cytokinesis defects and poly-ploidy as acute effects of CIN induction (Chunduri & Storchová, 2019; Mallin et al., 2022). Such cancer cell changes might explain why standard 2D phagocytosis assays show BMDMs attached to rigid plastic engulf relatively few anti-Tyrp1 opsonized cancer cells pretreated with MPS1i versus DMSO (Fig. S2D). In such cultures, BMDMs use their cytoskeleton to attach and spread, competing with engulfment of large and poorly opsonized targets. Noting that tumors in vivo are not as rigid as plastic, our 3D immunotumoroids eliminate attachment to plastic, and large numbers of macrophages can cluster and cooperate in engulfing cancer cells in a cohesive mass (Dooling et al., 2023). We indeed find CIN-afflicted tumoroids are eliminated by BMDMs regardless of anti-Tyrp1 opsonization (Fig. 2D-E), whereas anti-Tyrp1 is required for clearance of DMSO control tumoroids (Fig. 2D, S3B). Imaging also suggests that cancer CIN stimulates macrophages to cluster (compare Day-4 in Fig. 2D), which favors cooperative phagocytosis of tumoroids (Dooling et al., 2023), and occurs despite the lack of cancer cell opsonization and their larger cell size. The 3D immunotumoroid results with induced CIN are thus consistent with a more pro-phagocytic M1-type polarization (Fig.1J and 2B,C).

The authors used varying numbers of tumor cells for the in vivo portions of the study; the first half of the manuscript uses 500,000 cells, while the latter half uses 200,000 cells. Why?

The reasons for the difference in numbers is now clarified in the Methods:

For assessing immune infiltrates in early stages of tumor engraftment, when tumors are still small, we used a relatively high number of tumor cells (500,000 cells in Fig. 1D and Fig. 2F-G) to achieve sufficient cell numbers after dissociating the tumors, particularly for the slow-growing MPS1i-treated tumors. More specifically, with dissection, collagenase treatment, passage through a filter to remove clumps, we would lose many cells, and yet needed 100,000 viable cells or more for bulk RNA-seq suspensions and for flow cytometry measurements. For all other studies, 200,000 cancer cells were injected,

The authors need to report the tumor volumes and the total number of cells isolated from the day five tumors to avoid grossly inflating the effect (i.e. Fig 2G and 4G).

We have added relevant numbers in the Methods:

For day 5 post-challenge measurements, 100,000 to 200,000 live cells were collected. For in vivo tumor infiltrate studies in re-challenged mice, 10 million live cells were collected.

Also, regarding tumor sizes and cell numbers, we have previously published relevant measurements in assessments of tumor growth. Please see:

Brandon H Hayes, Hui Zhu, Jason C Andrechak, Lawrence J Dooling, Dennis E Discher, Titrating CD47 by mismatch CRISPR-interference reveals incomplete repression can eliminate IgG-opsonized tumors but limits induction of antitumor IgG, PNAS Nexus, Volume 2, Issue 8, August 2023, pgad243, https://doi.org/10.1093/pnasnexus/pgad243

Dooling, L.J., Andrechak, J.C., Hayes, B.H. et al. Cooperative phagocytosis of solid tumours by macrophages triggers durable anti-tumour responses. Nat. Biomed. Eng 7, 1081–1096 (2023). https://doi.org/10.1038/s41551-023-01031-3

In the present study, similar tumor growth curves are provided for transparency, but the Kaplan-Meier curves as the key pieces of data in Fig. 3-4. Lastly, regarding reporting total cell number harvested, we based our experiments on previously accepted measurements that also reported numbers out of total harvested cells. See:

Cerezo-Wallis, D., Contreras-Alcalde, M., … Soengas, M.S., 2020. Midkine rewires the melanoma microenvironment toward a tolerogenic and immune-resistant state. Nat Med 26, 1865–1877. https://doi.org/10.1038/s41591-020-1073-3

The figure titles need to be revised. For example, the title of Figure 1 claims that "MPS1i-induced chromosomal instability causes proliferation deficits in B16F10 tumors." However, the evidence provided is weak. The authors only present GSEA analysis of proliferation and no functional evidence of impairment. The authors need to characterize this proliferation deficit using in vitro studies and functional studies of macrophage polarization. I would suggest proliferation assays (crystal violet, MTT, Incucyte, etc) to measure the B16 growth over time with MPS1i treatment.

We thank the Reviewer for pointing this out. In Fig.1 we have minimized information regarding proliferation because it is later quantified in Figs.2D,E, S3, and 3D-i:

Fig.1F legend: Top downregulated hallmark gene sets in tumors comprised of MPS1i-treated B16F10 cells, showing downregulated DNA repair, cell cycle, and growth-related pathways, consistent with observations of slowed growth in culture and in vivo – as subsequently quantified.

Then the authors could collect the tumor supernatant to culture with macrophages and determine polarization in vitro. I would also like to see functional studies of macrophage polarization (suppression assays, cytokine production, etc). Currently, the authors provide no functional studies.

Fig.2B,C provides functional surface marker measurements of in vitro polarization toward anti-cancer M1 macrophages by MPS1i-pretreated tumor cells, consistent with gene expression in Fig.1G-J. Function is further shown as ant-cancer activity in Fig.2D,E, as now stated explicitly in the text:

“…In our 3D tumoroid in vitro assays, we found that macrophages can suppress the growth of chromosomally unstable tumoroids and clear them, surprisingly both with and without anti-Tyrp1 (Fig. 2D-E), regardless of MPS1i concentration used for treatment. Such a result is consistent with M1-type polarization (Fig.1J and 2B,C), which tends to be more pro-phagocytic. Such a result is consistent with M1-type polarization (Fig.1J and 2B,C), which tends to be more prophagocytic.”

The authors claim that macrophages are the key effector cells, but they need to provide evidence for this claim.

Other immune cells clearly contribute to the presented results because the IgG must eventually come from B cells. The text has been edited to indicate 'macrophages are key initiating-effector cells', and some evidence for this is the maximal survival of (WT B16 + Rev tumors) in Fig.3E upon treatment with Marrow Macrophages plus Macrophage-relevant SIRPa blockade and Macrophage-relevant IgG (via FcR). T cells do not have SIRPa or FcR.

They can deplete macrophages and T and B cells to determine whether the effect remains or is ablated. This is the only definitive way to make this claim.

To determine whether T and B cells might also be key initiating-effector cells, new experiments were done with mice depleted of T and B cells (per Fig.S9, below). We compared the growth of MPS1i vs DMSO treatments in these mice to results in mice with T and B cells (which should replicate our previous results in Fig.3D-i). We found that slower growth with Rev relative to DMSO was similar in mice without T and B cells compared to mice with T and B cells. We have added to the text our conclusion that: T and B cells are not key initiating-effector cells. Whereas B cells are effector cells at least in terms of eventually making anti-tumor IgG, our results show that macrophages are key initiating-effector cells because macrophages certainly affect the growth of (WT B16 + Rev tumors) when more are added (Fig.3E).

**Author response image 4. sa2fig4:** Growth of CIN-afflicted wild-type (WT) tumors in T- and B-cell deficient mice and T- and B-cell replete mice. Similar growth delays for MPS1i-pretreated B16F10 cells in T- and B-cell deficient NSG mice and immunocompetent C57BL/6 mice. Both types of mice have functional macrophages. Parallel studies in vivo were done with WT B16F10 ctrl cells cultured 24 h in 2.5 μM MPS1i reversine or DMSO, then washed 3x in growth media for 5 min each and allowed to recover in growth media for 48 h. 200,000 cells in 100 uL PBS were injected subcutaneously into right flanks, and the standard size limit was used to determine survival curves. The C57BL/6 experiments were done independently here (by co-author L.J.D.) from the similar results (by B.H.H.) shown in Fig.3D-i, which provides evidence of reproducibility.

The Results section final paragraph describes all of this:

Macrophages seem to be the key initiating-effector cells, based in part on the following findings. First, macrophages with both SIRPα blockade and FcR-engaging, tumor-targeting IgG maximize survival of mice with WT B16 + Rev tumors (Fig. 3E) – noting that macrophages but not T cells express SIRPα and FcR’s. Despite the clear benefits of adding macrophages, to further assess whether T and B cells are key initiating-effector cells, new experiments were done with mice depleted of T and B cells. We compared the growth delay of MPS1i versus DMSO treatments in these mice to the delay in fully immunocompetent mice with T and B cells – with all studies done at the same time. We found that slower growth with Rev relative to DMSO was similar in mice without T and B cells when compared to immunocompetent C57 mice (Fig.S9). We conclude therefore that T and B cells are not key initiating-effector cells. At later times, B cells are likely effector cells at least in terms of making anti-tumor IgG, and T cells in tumor re-challenges are also increased in number (Fig. 4G-ii). We further note that in our earlier collaborative study (Harding et al., 2017) WT B16 cells were pre-treated by genome-damaging irradiation before engraftment in C57 mice, and these cells grew minimally – similar to MPS1i treatment – while untreated WT B16 cells grew normally at a contralateral site in the same mouse. Such results indicate that T and B cells in C57BL/6 mice are not sufficiently stimulated by genome-damaged B16 cells to generically impact the growth of undamaged B16 cells.

**Reviewer #2 (Public Review):**
Harnessing macrophages to attack cancer is an immunotherapy strategy that has been steadily gaining interest. Whether macrophages alone can be powerful enough to permanently eliminate a tumor is a high-priority question. In addition, the factors making different tumors more vulnerable to macrophage attack have not been completely defined. In this paper, the authors find that chromosomal instability (CIN) in cancer cells improves the effect of macrophage targeted immunotherapies. They demonstrate that CIN tumors secrete factors that polarize macrophages to a more tumoricidal fate through several methods. The most compelling experiment is transferring conditioned media from MSP1 inhibited and control cancer cells, then using RNAseq to demonstrate that the MSP1-inhibited conditioned media causes a shift towards a more tumoricidal macrophage phenotype. In mice with MSP1 inhibited (CIN) B16 melanoma tumors, a combination of CD47 knockdown and anti-Tyrp1 IgG is sufficient for long term survival in nearly all mice. This combination is a striking improvement from conditions without CIN.Like any interesting paper, this study leaves several unanswered questions. First, how do CIN tumors repolarize macrophages? The authors demonstrate that conditioned media is sufficient for this repolarization, implicating secreted factors, but the specific mechanism is unclear. In addition, the connection between the broad, vaccination-like IgG response and CIN is not completely delineated. The authors demonstrate that mice who successfully clear CIN tumors have a broad anti-tumor IgG response. This broad IgG response has previously been demonstrated for tumors that do not have CIN. It is not clear if CIN specifically enhances the anti-tumor IgG response or if the broad IgG response is similar to other tumors. Finally, CIN is always induced with MSP1 inhibition. To specifically attribute this phenotype to CIN it would be most compelling to demonstrate that tumors with CIN unrelated to MSP1 inhibition are also able to repolarize macrophages.Overall, this is a thought-provoking study that will be of broad interest to many different fields including cancer biology, immunology and cell biology.

We thank the Reviewer for their enthusiastic and positive comments toward the manuscript.

Our main purpose with this study has been discovery science oriented and mechanistic, with implications for improving macrophage immunotherapies. More experimentation needs to be done to further understand how this positive immune response emerges. However, we could address whether CIN enhances or not the anti-tumor IgG response by quantitative comparisons to our two other recent studies, and we conclude that it does not per new edits in the Abstract and the Results. See attached PPT for full details and comparison.

Abstract:

“CIN does not greatly affect the level of the induced response but does significantly increase survival.”

“…these results demonstrate induction of a generally potent anti-cancer antibody response to CIN-afflicted B16F10 in a CD47 KO context. Importantly, comparing these sera results for CINafflicted tumors to our recent studies of the same tumor model without CIN (Dooling et al., 2022; Hayes et al., 2022), we find similar levels of IgG induction (e.g. ~100-fold above naive on average for IgG2a/c), similar increases in phagocytosis by sera opsonization (e.g. equivalent to antiTyrp1), and similar levels of suppressed tumoroid growth – including the variability.

…

However, median survival increased (21 days) compared to their naïve counterparts (14 days), supporting the initial hypothesis of prolonged survival and consistent not only with past results indicating major benefits of a prime-&-boost approach with anti-Tyrp1 (Dooling et al., 2022) but also with the noted similarities in induced IgG levels.”

Future studies could certainly focus on trying to identify what secreted factors might be inducing the M1-like polarization (using ELISA assays for cytokine detection, for example). This could be important because a main finding here is that we achieve nearly a 100% success rate in clearing tumors when we combine CD47 ablation and IgG opsonization with cancer cell CIN. Previous studies were only able to achieve about 40% cures in mice when working with CD47 disription and IgG opsonization alone, suggesting CIN in this experimental context does improve macrophage response.

Lastly, we agree with the Reviewer that future studies should also address how CIN in general (not MPS1i-induced) affects tumor growth. The final paragraph of our Discussion at least cites support for consistent effects of M1-like polarization:

“The effects of CIN and aneuploidy in macrophages certainly requires further investigation. We did publish recently that M1-like polarization of BMDMs with IFNg priming is sufficient to suppress growth of B16 tumoroids with anti-Tyrp1 opsonization more rapidly than unpolarized/unprimed macrophages and much more rapidly than M2-like polarization of BMDMs with IL4 (Extended Data Fig.5a in Dooling et al., 2023); hence, anti-cancer polarization contributes in this assay.

While the secretome from MPS1i-treated cancer cells has been found to trigger…”

Nonetheless, we can only speculate that there is a threshold of CIN reached by a certain timepoint in tumor engraftment and growth. Natural CIN might not be enough, so we pursued a pharmacological approach consistent with ongoing pre-clinical studies (https://doi.org/10.1158/1535-7163.MCT-15-0500). Future studies should consider trying knockdown models to gradually accrue CIN in tumors or using more relevant pharmacological drugs that are known to induce CIN not associated with the spindle. We believe, however, that these are larger questions on their own and are beyond the scope of the foundational discoveries in this manuscript.

**Reviewer #2 (Recommendations For The Authors):**
None

We again thank the Reviewer for their support and enthusiasm for the manuscript. We made some additional changes and more data to address questions posed by the other Reviewer that we hope you find to help the manuscript further.